# Timing of drought in the growing season and strong legacy effects determine the annual productivity of temperate grasses in a changing climate

Claudia Hahn[1], Andreas Lüscher[2], Sara Ernst-Hasler[1], Matthias Suter[2], Ansgar Kahmen[1]

[1]Department of Environmental Sciences - Botany, University of Basel, Schönbeinstrasse 6, CH-4056 Basel, Switzerland

[2]Forage Production and Grassland Systems, Agroscope, Reckenholzstrasse 191, CH-8046 Zurich, Switzerland

*Correspondance to:* Claudia Hahn (claudia.hahn@unibas.ch)

**Abstract**

The frequency of extreme weather events, such as droughts, is assumed to increase and lead to alterations of ecosystem productivity and thus the terrestrial carbon cycle. Although grasslands typically show reduced productivity in response to drought, the effects of drought on grassland productivity have been shown to vary strongly. Here we tested in a two-year field experiment, if the resistance and the recovery of grasses towards drought varies throughout a growing season and if the timing of drought influences drought-induced reductions in annual aboveground net primary production (ANPP) of grasses. For the experiment we grew six temperate and perennial C3 grass species and cultivars in a field as pure stands. The grasses were cut six times during the growing season and subject to 10-week drought treatments that occurred either in the spring, the summer or the fall. Averaged across all grasses, drought-induced losses of productivity in spring were smaller (-20% to -51%) than in summer and fall (-77% to -87%). This suggests a higher resistance to drought in spring when plants are in their reproductive stage and their productivity is the highest. After the release of drought, we found no prolonged suppression of growth. In contrast, post-drought growth rates of formerly drought-stressed swards outperformed the growth rates of the control swards. The strong overcompensation of growth after drought release resulted in relatively small overall drought-induced losses in annual ANPP that ranged from -4% to -14% and were not affected by the timing of the drought event. In summary, our results show that (i) the resistance of growth rates in grasses to drought varies across the season and is increased during the reproductive phenological stage when growth rates are highest, (ii) that positive legacy effects of drought indicate a high recovery potential of temperate grasses to drought, and (iii) that the high recovery can compensate immediate drought effects on total annual biomass production to a significant extent.

## 1. Introduction

Numerous studies have attempted to quantify the effects of drought on grassland ecosystems in the past decade. In general, these studies have confirmed that drought-induced water limitation typically leads to a reduction of net primary productivity (NPP) (Fuchslueger et al., 2014, 2016; Gherardi and Sala, 2019; Wilcox et al., 2017; Wu et al., 2011). Importantly, however, these studies have also shown that the response of ecosystems to experimental drought can vary quite dramatically (Gherardi and Sala, 2019; Gilgen and Buchmann, 2009; Grant et al., 2014; Hoover et al., 2014; Wilcox et al., 2017). Among others, the drought response of grasslands has been shown to depend on the severity of the experienced drought (Vicca et al., 2012; Wilcox et al., 2017), and important secondary factors such as the type of grassland affected (Byrne et al., 2013; Gherardi and Sala, 2019; Sala et al., 2015; Wilcox et al., 2017), the intensity of land use (Vogel et al., 2012; Walter et al., 2012), the plant functional composition (Gherardi and Sala, 2015; Hofer et al., 2016, 2017a; Mackie et al., 2018), or the biodiversity of an ecosystem (Haughey et al., 2018; Isbell et al., 2015; Kahmen et al., 2005; Wagg et al., 2017). These secondary factors that affect the responses of terrestrial ecosystems to drought are just beginning to be understood (Reichstein et al., 2013; Wu et al., 2011). Defining their impact on the drought response of terrestrial ecosystems is yet essential for quantitative predictions of drought effects on the carbon cycle and for the ultimate inclusion of drought responses of terrestrial ecosystems in coupled land surface models (Paschalis et al., 2020; Schiermeier, 2010; Smith et al., 2014).

Grassland ecosystems often show a pronounced seasonality, where plants undergo different phenological, physiological, morphological or ontogenetic stages throughout a year (Gibson, 2009; Voigtländer and Boeker, 1987). Temperate European grasslands for example, are highly productive early in the growing season during reproductive growth, while they show much lower growth rates during vegetative stages in summer and fall (Menzi et al., 1991; Voisin, 1988). Several studies have addressed how the seasonal timing of drought affects aboveground net primary productivity (ANPP) of North American C4 grasslands (Nippert et al., 2006; Petrie et al., 2018). It has been suggested that moisture availability during stalk production of the dominant C4 grass species in mid-summer is particularly important for maintaining the annual productivity of these grasslands (Denton et al., 2017; La Pierre et al., 2011). For C3 dominated temperate grasslands, this would imply that spring, when grasses flower and have the highest growth rates, is the time when the productivity should be most susceptible to drought and that productivity should be less prone to drought-induced losses in the summer and fall. Empirical evidence how the seasonal timing of a drought event affects the productivity of temperate C3 dominated grasslands is, however, missing.

The impact of drought on the annual ANPP of ecosystems depends on the immediate effects of drought on productivity (determined by the drought resistance of the ecosystem), but also on potential legacy effects that occur after drought release (determined by the drought recovery of the ecosystem) (Sala et al., 2012; Seastedt and Knapp, 1993). In particular legacy effects of drought are a critical yet rarely explored component that can strongly affect the impact of

drought on the annual ANPP of an ecosystem (Finn et al., 2018; Ingrisch and Bahn, 2018; Petrie et al., 2018; Sala et al., 2012). Previously it was believed that the drought history (e.g. previous year annual precipitation deficit) of an ecosystem is crucial for the annual ANPP and that the magnitude of the drought history negatively influences current ANPP (Mackie et al., 2018; Reichmann et al., 2013; Sala et al., 2012; Yahdjian and Sala, 2006). In contrast, there is now increasing evidence that drought stressed plants or ecosystems can respond to drought release also with an overcompensation of their physiological activity or growth (Griffin-Nolan et al., 2018; Hofer et al., 2017a; Shen et al., 2016). Following an experimental drought, tropical and temperate tree seedlings have, for example, exhibited higher net photosynthesis rates than seedlings that had not experienced a drought event (Hagedorn et al., 2016; O'Brien et al., 2017). In grasslands, Hofer et al. (2016) recently have shown that formerly drought-stressed swards had a higher productivity in the post-drought period than non-stressed control swards. Other studies have shown that the species richness of a grassland contributes to this effect (Kreyling et al., 2017; Wagg et al., 2017). Even across growing seasons it has been suggested that the previous growing season precipitation patterns can have positive legacy effects on the current year productivity of ecosystems (Shen et al., 2016). As legacy effects can either worsen or diminish immediate drought effects on annual ANPP, their assessment is essential to determine if the sensitivity of annual ANPP to the timing of drought is driven by the resistance or the recovery of the system (Petrie et al., 2018; Shen et al., 2016). This requires, however, a detailed analysis of not only annual ANPP, but the assessment of biomass increase (i.e. productivity) during and after the release of a drought event.

In the work that we present here, we experimentally assessed if the drought response of the annual ANPP (i.d. the productivity of standing above-ground biomass) of six different grass species and cultivars that are common in temperate C3 grasslands depends on the timing of the drought event in the growing season. To do so, we determined the drought resistance and recovery for these grasses in different times of the growing season. Specifically, we tested in our study,

i) if the timing of a drought event within the growing season (e.g. spring, summer, fall) has an effect on the immediate aboveground productivity reduction – i.e. the resistance of an ecosystem,

ii) if the timing of a drought event within the growing season affects the recovery of an ecosystem, and

iii) how the combination of resistance and recovery in different times of the growing season impacts the annual ANPP of drought-stressed C3 grasses.

## 2. Materials and methods

### 2.1 Research site

The experiment was performed in the years 2014 and 2015 near Zurich, Switzerland (47°26'N, 8°31'E, altitude: 490 m a.s.l., mean annual temperature: 9.4°C, mean annual precipitation: 1031 mm) on an eutric cambisol soil. For the experiment, we established four perennial C3 grass species, two of them in two cultivars, all of which are commonly used in agricultural practice in August 2013 on 96 plots (3 × 5 m). The grasses were sown as pure stands on a highly productive field that yields typically around 12 t grass dry matter per year and hectare (i.e. 1200 g m$^{-2}$). The establishment followed the basic procedures of sowing permanent highly productive grasslands, where before sowing, the existing vegetation at the site (which was a winter wheat) was plowed. The grasses were established in the growing season before the experiment started following best practice which guaranteed full establishment of the swards (including vernalisation during winter) and full productivity in the following year. The six grasses were *Lolium perenne* L. early flowering (LPe; cultivar 'Artesia'), *Lolium perenne* L. late flowering (LPl; cultivar 'Elgon'), *Dactylis glomerata* L. early flowering (DGe; cultivar 'Barexcel'), *Dactylis glomerata* L. late flowering (DGl; cultivar 'Beluga'), *Lolium multiflorum* Lam. var *italicum* Beck (LM; cultivar 'Midas'), and *Poa pratensis* L. (PP; cultivar 'Lato'). Phosphorous, potassium and manganese were applied following national Swiss fertilization recommendations for intensely managed grasslands at the beginning of each growing season (39 kg P ha$^{-1}$, 228 kg K ha$^{-1}$, 35 kg Mg ha$^{-1}$). In addition, all plots received the same amount of mineral N fertilizer as ammonium-nitrate (280 kg N ha$^{-1}$, divided into six applications per year). The solid N fertilizer was applied at the beginning of the growing season (80 kg N ha$^{-1}$) and after each of the first five cuts (40 kg N ha$^{-1}$ each time).

### 2.2 Experimental design

Each of the six grass species (different species and cultivars) was subject to four treatments: one rain-fed control and three seasonal drought treatments (spring, summer, fall) (see Fig. 1). We used a randomized complete block design with four blocks representing the four replicates. Each block contained all the 24 plots (six species times four treatments) fully randomized. A drought treatment lasted for ten weeks. Drought was simulated using rainout shelters that excluded rainfall completely on the treatment plots. The rainout shelters were tunnel-shaped and consisted of steel frames (3 × 5.5 m, height: 140 cm) that were covered with transparent and UV radiation transmissible greenhouse foil (Lumisol clear, 200 my, Hortuna AG, Winikon, Switzerland). To allow air circulation, shelters were open on both opposing short ends and had ventilation openings of 35 cm height over the entire length at the top and the bottom at both long sides. Gutters were installed to prevent the water from flowing onto adjacent plots, and a 0.75 m boarder zone at each plot was not considered for measurements to prevent a possible effect of lateral water flow in the soil. These shelters and plot design had previously been successfully used in other grassland-drought experiments (Hofer et al., 2016, 2017a, 2017b). Rain-fed controls were subject to the natural precipitation regime. However, when soil water potential ($\Psi_{Soil}$) sank below -0.5 MPa

due to naturally dry conditions, control plots were additionally watered with 20 mm of water (300 l per plot). In summer 2014 the irrigation was delayed by approximately one week due to organizational difficulties, leading to a further decrease in $\Psi_{Soil}$ until irrigation could start. Watering happened once on June 16[th] and 17[th] 2014 and three times in 2015 (7.7., 14.7., 11.8.).

### 2.3 Environmental measurements

Relative humidity and air temperature were measured hourly at the field site using VP-3 humidity, temperature and vapor pressure sensors (Decagon Devices, Inc., Pullman, WA, USA). Measurements were conducted in control and treatment plots under the rainout shelters (n=2). Information on precipitation and evapotranspiration was provided by the national meteorological service stations (MeteoSchweiz) that were in close proximity of our research site (average of the two surrounding meteorological stations Zurich Affoltern in 1.4 km distance and Zurich Kloten in 4.5 km distance). $\Psi_{Soil}$ was measured in 10 cm depth on an hourly basis using 32 MPS-2 dielectric water potential sensors (Decagon Devices, Inc., Pullman, WA, USA). The 32 soil water potential sensors were evenly distributed over the field and treatments. Daily means of all measurements were calculated per treatment, but across grasses since no grass-specific alterations in $\Psi_{Soil}$ were expected (Hoekstra et al., 2014) or measured (n=8).

In addition to soil water potential, we determined the stress intensity ($I_S$) as a metric to compare plant responses to reduced water availability (Vicca et al., 2012). It reflects the actual treatment experienced by plants. $I_s$ was calculated as in Granier et al. (2007):

$$I_S = sum(max[0, (TH-REW_t)/TH]). \qquad \text{Eq. (1)}$$

Where TH is the threshold (i.e. TH = 0.4; Granier et al. (2007)) and $REW_t$ is the relative extractable soil water on day t. REW is calculated as follows (Jiao et al., 2019):

$$REW = (\Psi_{Soil} - \Psi_{Soil\ wp}) / (\Psi_{Soil\ fc} - \Psi_{Soil\ wp}), \qquad \text{Eg (2)}$$

with $\Psi_{Soil\ wp}$ being the soil water potential at field capacity (i.e. -0.03 MPa; Granier et al. (2007)) and $\Psi_{Soil\ fc}$ being the soil water potential at wilting point (-1.5 MPa).

### 2.4 Harvests

Aboveground biomass was harvested six times per year in five-week intervals in 2014 and 2015, resulting in six growth periods per year (see Fig. 1). Aboveground biomass was also harvested once in spring 2016. Such a high frequency of harvests is typical for highly productive European grasslands used for fodder production. For the purpose of our study this high-resolution biomass sampling allows the analyses of the immediate drought effects and the impacts of drought that occur after the release of drought on productivity. The harvests were synchronized with the drought treatments and occurred five and ten weeks after the installation of the shelters on a respective treatment. For the harvest, aboveground biomass was cut at 7 cm height above the ground and harvested from a central strip (5 × 1.5 m) of the plot (5 × 3 m) using an experimental plot harvester (Hege 212, Wintersteiger AG, Ried/I., Austria). The fresh weight of the total harvest of a plot was determined with an integrated balance directly on the plot harvester. Dry biomass production was determined by assessing dry weight – fresh weight ratios of the harvested biomass. For this a biomass subsample was collected for each plot and the fresh and dry weight (dried at 60°C for 48 h) were determined. After the harvest of the aboveground biomass in the central strip of a plot, the remaining standing biomass in a plot was mowed 7 cm above the ground and removed.

### 2.5    Roots

Belowground biomass of four grasses (DGe, DGl, LPe and LPl) was harvested six times per year. For each treatment samples were collected at the end of a drought treatment and six to eight weeks after drought release from the respective treatment and control plots. Samples were collected using a manual soil auger with a diameter of 7 cm. For each plot samples of the upper 14 cm soil were taken from two different spots (one sample directly from a tussock and one from in between tussocks) and pooled as one sample per plot. All samples were washed using a sieve with a mesh size of 0.5 cm × 0.5 cm and weighed after drying (at 60°C for 72 h).

### 2.6    Determining drought impacts on productivity

In order to allow the comparison of grassland productivity in the different treatments across the two years we standardized the productivity that occurred in between two harvest periods (i.e. during five weeks) for growth related temperature effects and calculated temperature-weighted growth rates for each of the six grasses (DMYTsum, see Menzi et al. (1991)). For this purpose, we determined temperature sums of daily mean air temperature (as measured in the treatment and control plots) above a baseline temperature of 5°C (Tsum) for each growth period (i.e. 5 weeks prior to harvest). Dry matter yield (DMY) of a given harvest was then divided by the temperature sum of the corresponding time period to obtain temperature-weighted growth rates (henceforth referred to simple as growth rate):

$$DMYTsum = DMY(g\ m^{-2})/Tsum(°C). \qquad\qquad Eq.\ (3)$$

To determine the absolute change of growth (ACG) of a drought treatment on aboveground growth rate we calculated the
difference between temperature-weighted growth rates in a drought treatment (drt) and the corresponding control (ctr):
$\qquad$ ACG = DMYTsum(drt)-DMYTsum(ctr). $\qquad$ Eq. (4)
To determine the relative change of growth (RCG) due to drought, we calculated percentage change of temperature-
weighted growth rates:
$\qquad$ RCG = 100×(DMYTsum(drt)/DMYTsum(ctr)-1). $\qquad$ Eq. (5)
Annual ANPP as an average of the different grasses was determined by adding up the dry matter yields of the six harvests
of a growing season. These data were not temperature-corrected dry matter yield (DMY).

### 2.7 Data analysis

Relative and absolute changes in DMYTsum due to drought, the season of drought, and the tested grasses were analyzed
using linear mixed-effects models (Pinheiro and Bates, 2000). Temperature-weighted growth rate (DMYTsum) was
regressed on the fixed variables season (factor of three levels: spring, summer, fall), drought (factor of two levels: control,
drought treatment) and grass (factor of six levels: LPe, LPl, DGe, DGl, LM, PP), including all interactions. To account
for repeated measurements of the control plots over time (as the control for every seasonal drought treatment was the
same), plot was specified as a random factor, thereby accounting for potential correlation of DMYTsum over time.
DMYTsum was natural log-transformed prior to analysis to improve homogeneity and normal distribution of residual
variance. This transformation also implies that the regressions provide the inference to relative changes in DMYTsum,
namely RCG. A temporal compound symmetry correlation structure was initially imposed on the residuals, yet, it turned
out that the estimated correlation parameter was very small. A likelihood ratio test indicated its non-significance ($p>0.5$)
and it was finally omitted. However, inspection of residuals revealed clear differences in their variance among seasons
and control and drought plots, and the residual variance parameter was defined as $\mathrm{Var}(e_{jk}) = \sigma^2 \delta_{jk}^2$, with $\delta$ being a ratio to
represent $j \times k$ variances, one for each of three seasons $j$ under control and drought conditions $k$ (Pinheiro and Bates,
2000). The marginal and conditional $R^2$ of the model was calculated following Nakagawa and Schielzeth (2013). This
model was applied to DMYTsum at each second growth period under drought and the second post-drought growth period
in 2014 and 2015. Finally, absolute changes in DMYTsum are displayed in figures to improve the interpretation of the
data.
Root dry weight was analyzed in a similar way, i.e. it was natural log-transformed prior to analyses and the same
explanatory factors were applied in a mixed model, except that the factor grass had only four levels (only LPe, LPl, DGe
and DGl measured). Here, estimation of a single residual variance parameter $e_i$ was sufficient to fulfill the model
assumptions. This model was applied to root dry weight harvested in 2014 at the end of each drought treatment and six
to eight weeks after drought-release. Absolute changes in root dry weight are displayed in figures without further tests.
Annual ANPP was analyzed by two-way analysis of variance (ANOVA). The first factor season-treatment
consisted of the four levels control, spring drought, summer drought, and fall drought; the second factor grass consisted
of six levels, representing the six grasses. This ANOVA was performed for each of the years 2014 and 2015.
All statistical analyses were done using the statistical software R, version 3.5.1 (R Foundation for Statistical
Computing, Vienna, Austria, 2018). Mixed-effects models were fit using the package *nlme*, version 3.1-137, (Pinheiro
and Bates, 2000), and graphics were implemented with the package *ggplot2*, version 2.1.0 (Wickham, 2016).

## 3. Results

### 3.1 Precipitation, evapotranspiration and soil water potential

Over the entire growing season, the year 2015 was exceptionally dry, while 2014 showed normal weather conditions for the experimental site. The difference between rainfall (717.9 and 648.5 mm for 2014 and 2015, respectively; see Tab. 1) and evapotranspiration (356 and 447 mm for 2014 and 2015, respectively; shown in Fig. 1), i.e. the ecosystem water balance, was 361.9 mm in 2014 and only 201.5 mm in 2015 for the unsheltered control plots. The shelter periods reduced the total annual precipitation in the different treatments between -17.9 % and -37.0 % and the precipitation of the growing season (duration of the experiment, approx. March – November) by between -23.1 % and -45.8 % (see Table 1).

In 2014 $\Psi_{Soil}$ was severely reduced in the drought treatments and reached values around the permanent wilting point (-1.5 MPa) for the entire second half of the sheltered periods in all treatments (spring, summer, fall) (Fig. 2b-e, Table 2). Due to low rainfall in June 2014, $\Psi_{Soil}$ dropped not only in the sheltered summer drought treatment, but also in the control and the fall drought treatment (that was not yet sheltered). $\Psi_{Soil}$ recovered in the treatment plots after each sheltered period and reached $\Psi_{Soil}$ values comparable to the ones in the control plots. Because of the lack of rain in June 2014, the full rewetting of the spring drought treatment occurred only in the second post-drought growth period after the spring drought shelter period, while after the summer drought treatment rewetting occurred already in the first post-drought growth period.

In 2015, drought treatments reduced $\Psi_{Soil}$ in all seasons (Fig. 2g-k). However, an intense rain event caused some surface runoff in the field on May 1st 2015, which partly interrupted the spring drought treatment. Still, for the second growth period of the spring drought treatment of 2015 the median of $\Psi_{Soil}$ was at -0.77 MPa, a value comparable to that of the second growth period of the summer drought treatment (-0.83 MPa) (Table 2). Also Is values demonstrate that water stress severity in weeks six to 10 of the spring treatment (Is=14) reached levels at least as severe as during the corresponding weeks of the summer drought treatment (Is=4; Table 2). In 2015 $\Psi_{Soil}$ reached lower values during the shelter period in the fall treatment than during the shelter period in the spring and summer treatments. Due to a lack of rain in 2015, $\Psi_{Soil}$ and Is values recovered only partly after the end of the shelter period in the spring and summer drought treatments and remained significantly below that of the control plots for both post-drought growth periods (Table 2). Watering of the control plots during natural dry conditions lead to quick increases in $\Psi_{Soil}$ to values close to saturation (=0 MPa).

Daily mean air temperature under the rainout shelters was 0.7°C and 0.6°C higher in 2014 and 2015, respectively (Table 2).

### 3.2 Varying growth rates throughout the growing season

The temperature-weighted growth rates of the investigated six grass species and cultivars in the control plots showed a very strong seasonal pattern (Fig. 3a). In both years, it was highest during the second growth period in spring and sharply declined to values that were two- to eight-fold smaller in summer and fall. In summer and autumn 2015 growth rates of the grasses were clearly lower than in 2014. Root biomass increased towards summer and slightly decreased after summer in 2014 (Fig. 3b, Table A1; season $p<0.001$).

### 3.3 Seasonality of drought resistance

The growth rates of the six grass species and cultivars were barely affected by the exclusion of rain during the first five weeks of sheltering (Fig. 4). However, during the second sheltered growth period (drought weeks six to ten), the drought treatments strongly reduced temperature-weighted growth rates in all seasons, in both years, and in relative and absolute terms (Figs. 4, and 5, Table 3). In both years, averaged over all six grasses, the relative drought-induced changes in growth rates compared to the controls were smallest in spring (2014: -51%, 2015: -20%) and clearly larger in summer (2014: -81%, 2015: -85%) and fall (2014: -77%, 2015: - 84%) (Fig.4a, Table 3; season x treatment $p<0.001$). As such, the drought resistance of temperate grasses throughout the growing season was largest in spring when their growth rates in the control were especially high (Fig. 3a; second regrowth). This pattern was generally observed for all six grass species and cultivars tested (Fig. 5a) even though there was a significant season × treatment × grass interaction (Table 3). In 2014 this interaction mainly derived from DGl and PP showing an exceptionally large drought induced growth reduction in fall. In 2015 it was explained by an especially low drought response of DGl in spring and strong responses of DGl in summer and LPe and PP in fall (Fig. 5a).

In 2014 the absolute drought-induced reduction of growth across all six grass species and cultivars was largest in spring (-0.5 g m$^{-2}$ °C$^{-1}$), followed by summer (-0.4 g m$^{-2}$ °C$^{-1}$) and was lowest in the fall (-0.1 g m$^{-2}$ °C$^{-1}$) (Fig. 4b). Likewise, in 2015 the absolute reduction of the growth rate in the drought treated plots was largest across the six grass species and cultivars in spring (-0.2 g m$^{-2}$ °C$^{-1}$), but slightly lower in summer (-0.1 g m$^{-2}$ °C$^{-1}$) and fall (-0.1 g m$^{-2}$ °C$^{-1}$).

The average standing root biomass across four of the grasses was not significantly affected by any of the drought treatments of 2014 (Fig. 6, Table A1; treatment $p=0.572$, season x treatment $p=0.825$).

### 3.4 Seasonality of post-drought recovery

When compared to corresponding controls, relative and absolute changes in temperature-weighted growth rates after drought release showed positive treatment effects in 2014 (Fig. 7, Table 4). Across all six grass species and cultivars, the relative increases in post-drought growth rates were 41% after the spring drought treatment, 31% after the summer drought treatment and 53% after the fall drought treatment, and did not differ among the seasons (Table 4; season × treatment $p=0.180$). In 2015, the relative increases in post-drought growth rates were 5% after the spring drought treatment, 15%

after the summer drought treatment and 52% after the fall drought treatment, and did differ among the seasons (Table 4; season × treatment $p<0.001$). Increased relative and absolute growth rates were also observed in the first harvest in 2015 and 2016 for all the plots that had received a drought treatment in 2014 and 2015, respectively (Fig. 4). In this first harvest of 2015, relative growth rate increases were 110% after the spring, 36% after the summer and 53% after the fall drought treatments of 2014. In the first harvest of 2016, relative growth rate increases were 10% after the spring, 31% after the summer and 51% after the fall drought treatments of 2015.

When compared across the different grass species and cultivars, the only grass that tended to have a weak recovery (lower or no increase of growth rate during post-drought) was LM (Fig. 7); but there was no significant difference among the grass species and cultivars (Table 4; treatment x grass $p=0.517$). In 2015 again LM showed the weakest recovery of all the grasses after all drought treatments, the effect being significant (Table 4; treatment x grass $p<0.001$).

Root dry weight of the treatment plants generally showed no alterations in growth compared to the control (Fig. 6, Table A1; treatment $p=0.553$).

### 3.5    *Effects of seasonal drought on annual biomass production*

The cumulative annual aboveground biomass production (annual ANPP) of the controls averaged across all six grass species and cultivars differed strongly between the two years (Fig. 8a), with 2014 (1303 g m$^{-2}$ a$^{-1}$) being 37% more productive than 2015 (949 g m$^{-2}$ a$^{-1}$). The strong reduction in biomass production in 2015 was probably related to the naturally occurring lack of rain in summer and fall (Fig 2). But because the control was irrigated when strong stress occurred this cannot explain the whole extent. This is evident from the two spring growth periods being equally productive in the unsheltered plots (control, summer and fall drought) in 2015 and in 2014 (Fig. 8). The annual ANPP of the treatments was significantly different from control in both years (Table A2; season-treatment $p<0.001$ for 2014 and $p=0.007$ for 2015). In 2014, the largest drought effect on the annual ANPP across all grasses resulted from the summer treatment, which reduced productivity significantly by -14% (185 g m$^{-2}$) compared to the control (Fig 8). Spring and fall drought treatments in 2014 resulted in a non-significant -4% (-53 g m$^{-2}$) and -6% (-74 g m$^{-2}$) reduction of annual ANPP across all grass species and cultivars, respectively. In 2015, drought treatments in the summer and fall significantly caused a -10% and -11% reduction of annual ANPP across all grasses (-97 g m$^{-2}$ and -105 g m$^{-2}$), respectively, while the spring drought treatment reduced annual ANPP across all grasses by only -4% (-34 g m$^{-2}$), which was not significant (Fig. 8).

## 4. Discussion

In our study we experimentally assessed if the drought resistance and recovery of six different temperate perennial C3 grass species and cultivars varies throughout the growing season and if the timing of a drought event has an influence on drought induced reductions in annual ANPP of these grasses. All six temperate grass species and cultivars showed a clear seasonal pattern of drought resistance in both years. The drought-induced reduction of growth was smaller under spring drought (-20% and -51% for the two years when averaged across the six grasses) than under summer and fall droughts (between -77% and -87%). Thus, the investigated grasslands were more resistant to drought in the spring when productivity of temperate grasses is generally the highest and they were least resistant in summer and fall, when their productivity is much lower. Moreover, the examined grasslands did not show any negative legacy effects such as a prolonged suppression of growth after rewetting following the end of the drought treatments. In contrast, after the release of drought, temperature-weighted growth rates of the grasses in the treatment plots surprisingly outperformed the growth rates of the grasses in the controls for extended periods of time. This suggests a high recovery potential of all six grasses that we investigated. As a consequence of the high recovery, the seasonal drought treatments resulted in only moderate drought-induced reductions in annual ANPP between -4% to -14% - despite the strong immediate effects of drought - and no clear effects of the timing of drought on annual ANPP were detected. With this our study shows (i) that the resistance of growth rates in different grasses to drought varies throughout the growing season and is increased during the reproductive phenological stage when growth rates in the control were highest, (ii) that positive legacy effects of drought on plant productivity indicate a high recovery potential of temperate C3 grasses throughout the entire growing season, and (iii) that the high recovery can compensate to a significant extent for immediate seasonal drought effects on productivity, resulting in total annual ANPP that is only marginally reduced in the drought treated plots compared to the controls.

### 4.1 Differences in the meteorological conditions between the two years

While the first experimental year (2014) was characterized by more or less normal meteorological and thus growth conditions, the summer of 2015 was exceptionally dry in all of central Europe (Dietrich et al., 2018; Orth et al., 2016). These conditions led to a reduction of the annual ANPP of the control plots by -37% in 2015 compared to 2014 (Fig. 8). The lack of precipitation in the second half of the 2015 growing season, i.e. between the third harvest in June and the last harvest in October (Fig. 2), was of importance for our experiment, especially for the response of the treatments during the recovery phase after the removal of the shelters. In this period, the amount of rainfall was only 153 mm in 2015 while it was 405 mm in 2014. Thus, positive legacy effects directly following drought treatments were much smaller or absent following the spring and summer treatments in 2015 due to a missing rewetting (Figs. 2, 4 and 7). Yet, strong positive legacy effects in response to the 2015 treatments were observed in the first harvest of 2016 when the experimental site

was fully rehydrated. This highlights the general occurrence of positive drought legacy effects in the investigated grasslands once the soil moisture has recovered from the drought treatments and indicates some long-lasting mechanisms behind this overcompensation, as full rewetting occurred already half a year before the harvest in 2016.

Intense rains between the first and second harvest of the year 2015 caused some water flow into the treatments. This resulted in a partial reduction of drought stress in the treatment plots (Fig. 2h). Yet, both the median of the soil water potential and the Is were still clearly reduced in the treatment plots compared to the control and, consequently, we observed a reduction of growth rates in the second spring harvest in 2015 despite this event (Figs. 4, 5). We therefore conclude that the partial reduction in drought stress did weaken the immediate drought response during the growth period concerned, but that this does not question the overall drought responses of the grasslands that we report here. This is especially evident from the drought stress during weeks six to ten being of comparable severity (Table 2).

### 4.2 *Grasses were most resistant to drought in spring, the most productive phenological stage*

Previous studies have indicated that the timing of drought is relevant for the reduction of annual ANPP of ecosystems (Bates et al., 2006; Denton et al., 2017; La Pierre et al., 2011; Nippert et al., 2006). It has been argued that the variable drought sensitivity of ecosystems throughout the growing season could be linked to different phenological stages of dominant plant species, where plants in reproductive stages and periods of high growth are particularly susceptible to drought (Bates et al., 2006; Craine et al., 2012; Dietrich and Smith, 2016; Heitschmidt and Vermeire, 2006; O'Toole, 1982). We found, however, that relative reductions in temperature-weighted growth rates were lowest in the spring treatments 2014 and 2015 as compared to the summer and fall treatments. The highest resistance of plant growth rates to drought occurred, thus, when the plants showed the highest growth rates in the control (Fig. 3) and when the investigated grasses were in their reproductive stages. With this, our findings are in contrast to previous studies that have suggested temperate grasslands and crops to be particularly susceptible to drought early in the growing season when their growth rates are the highest and plants are in reproductive stages (Bates et al., 2006; Craine et al., 2012; Dietrich and Smith, 2016; Heitschmidt and Vermeire, 2006; Jongen et al., 2011; O'Toole, 1982; Robertson et al., 2009). Our study does support, however, findings of El Hafid et al. (1998) and Simane et al. (1993), who detected that spring droughts have the least impact on annual productivity of wheat. Importantly, most of the previous studies that have reported the effects of drought timing on grasslands or other ecosystems report effects on annual ANPP but have not differentiated between immediate effects and legacy effects of drought events as we did in our study. As drought impacts on annual ANPP combine immediate and post drought legacy effects, it is difficult to directly compare the results we present here on variably seasonal drought resistance of temperate C3 grasses to previous work reporting the influence of drought timing on annual ANPP.

One possibility for the higher drought resistance of grasses during spring is that grasses invest more resources towards the stress resistance of their tissue in this part of the growing season when they have not only the largest growth rates, but also reproduce. Such a resource allocation strategy could allow drought stressed grasses to remain physiologically active in this critical part of the growing season. Osmotic adjustment is one mechanism that reduces the effects of drought on the physiological performance of the plant (Sanders and Arndt, 2012). This is achieved through the active accumulation of organic and inorganic solutes within the plant cell. Thus, osmotic potential increases and the plant can withstand more negative water potentials in the cell while maintaining its hydraulic integrity (Sánchez et al., 1998). Santamaria et al. (1990) found that early- and late flowering cultivars of *Sorghum bicolor* L. developed a different pattern of osmotic adjustment (continuous increase of osmotic adjustment vs. first increase and later decrease of osmotic adjustment), hinting that drought tolerance may vary among seasons. In a companion paper we report physiological data for the six grasses from the same experiment. We show that at a given soil water potential, foliar water potentials were less negative and stomatal conductance was higher in plants drought stressed in the spring compared to plants drought stressed in the summer or fall (Hahn et al. in prep). This suggests indeed that for a given drought level, grasses remain physiologically more active in the spring than in the summer or fall. The exact physiological mechanisms that explain the higher drought resistance of the investigated grasslands in the spring and their higher drought susceptibility in the summer and fall remain yet unknown and require further detailed ecophysiological and biochemical assessments.

An alternative explanation for different immediate drought effects on growth rates throughout the growing season are different experimentally induced drought severities throughout a growing season. This could be by either residual moisture from winter dampening the experimentally induced drought more in the spring than in the summer or fall. Alternatively, higher evaporative demand of the atmosphere in the summer compared to the spring or fall could have enhanced experimentally induced drought effects in the summer. De Boeck et al. (2011) explain for example the higher drought susceptibility of growth in three herbs in the summer compared to spring by a higher evaporative demand of the atmosphere in the summer compared to spring or fall. In our study, however, soil water potential data as well as drought stress intensity Is indicate that ten weeks of drought treatment resulted in mostly equal water depletion and stress levels in spring, summer and fall (Fig. 2, Table 2). In addition, we found only small differences in median VPD between the spring, summer and fall drought treatment period (Fig. 2). This suggests that stronger drought stress in summer and fall compared to spring cannot explain alone the different resistances of plant growth to drought throughout the growing season. Along these lines, Denton et al. (2017), who performed a similar experiment as we report here but in a C4 grassland in North America, also did not find that these seasonal differences in the experimentally induced drought severity are the reason for variable drought effects on the growth rates throughout the growing season.

### 4.3 *No increased root biomass in the top soil layer*

In the entire experiment, root biomass did not generally increase under drought (Table A1), and only increased in one of the investigated grasses (DGe) in one (summer) of the three treatments. This confirms the findings of Byrne et al. (2013), Denton et al. (2017) and Gill et al. (2002), who did not find any changes in belowground biomass in response to drought. In a similar setting, Gilgen and Buchmann (2009) found no changes in belowground biomass to simulated summer drought in three different temperate grassland sites (from lowland to alpine grassland). While Denton et al. (2017) ascribe the missing drought response in belowground biomass to modest precipitation alterations in their experiment, we can exclude this factor in our experiment since the soil water potential under drought was significantly reduced compared to the soil water potential in the controls in every season. Contrary to our finding, several studies have shown that drought can maintain or increase root growth while inhibiting shoot growth (Davies and Zhang, 1991; Hofer et al., 2017a; Saab et al., 1990). In an experiment by Jupp and Newman (1987), *L. perenne* increased lateral root growth under low $\Psi_{Soil}$ indicating an increased investment in root growth under water limited conditions. In our experiment the *L. perenne* grasses did not show a trend towards increased investment in root growth, neither during drought nor after drought-release, contradicting the results of Jupp and Newman (1987). Such differences in the response of root biomass in different studies as described above may derive from the soil layer that was investigated. Hofer et al. (2017a) have shown that the response of root growth into ingrowth bags depended on the soil depth: root growth of *L. perenne* decreased in the top soil layer (0-10 cm), but increased in deeper soil layers of 10-30 cm. Thus, the superficial root sampling (0-14 cm) in our experiment might mask increased root growth in deeper soil layers.

### 4.4 Positive legacy effects of drought periods

Several previous studies have suggested that drought events can lead to negative legacy effects on the productivity of ecosystems (De Boeck et al., 2018; Petrie et al., 2018; Reichmann et al., 2013; Sala et al., 2012). We found, however, that growth rates of previously drought-stressed plots were significantly larger than in the corresponding control plots after rewetting, indicating not only a high recovery potential of the investigated grasses but even positive legacy effects (Figs. 4 and 7). Interestingly, we did not only observe growth rates that were larger in the treatment plots than in the control plots immediately after the drought release, but observed larger growth rates in all treatment plots compared to the control plots even in the first harvests of the following growing season (Fig. 4). This pattern was consistent for both years of the experiment. Bloor and Bardgett (2012) and also Denton et al. (2017) found that drought events promote soil fertility and nutrient retention following drought release. Likewise, Gordon et al. (2008) found an increase in microbial activity after a rewetting event, possibly leading to a rapid and sudden influx of plant available nutrients in the soil (Mackie et al., 2018; Schimel and Bennett, 2004; Van Sundert et al., 2020). Hofer et al. (2017a) also attributed growth increases relative to control plots in post-drought periods to nitrogen availability in the soil and Karlowsky et al. (2018) found evidence that interactions between plants and microbes increase plant nitrogen uptake in grasslands after rewetting

445 events. It could, thus, be that the enhanced productivity in the treatment plots following drought release is the result of

446 increased microbial activity leading to enhanced nitrogen availability and/or changes in resource limitation following

447 drought release as suggested by Seastedt and Knapp (1993) in their Transient Maxima Hypothesis.

448   We applied nitrogen fertilizer in our experiment to each plot after each harvest, also at the beginning and in the

449 middle of a drought treatment. Since we applied the fertilizer in form of water-soluble pellets, it is possible that

450 precipitation exclusion prevented dissolution and, thus, nitrogen fertilizer pellets could have accumulated in the drought-

451 treated plots during the treatment phase. The rewetting of the soil could have resulted in a massive release of nitrogen

452 fertilizer from these pellets so that plant growth rates in formerly drought-stressed plots were stimulated by the release of

453 this fertilizer and, thus, was larger than those of the control plots. However, Hofer et al. (2017a) observed strongly

454 increased N availability and plant growth rates after drought release not only in plots that received mineral fertilizer during

455 the drought treatment period, but also in plots that did not receive any N fertilizer during drought. We suggest therefore

456 that the release of accumulated fertilizer nitrogen in the treatment plots might explain some, but not all post-treatment

457 growth responses in the formerly drought treated plots in our study.

458   Hagedorn et al. (2016) have shown that rewetting events trigger intrinsic processes that lead to a sudden increase

459 of photosynthesis in young beech trees. Moreover, Arend et al. (2016) found a rapid stimulation of photosynthesis

460 immediately after rewetting that continued until the end of the growing season, partly compensating the loss of

461 photosynthetic activity during drought. Hofer et al. (2017b) found an increased root mass and increased water-soluble

462 carbohydrate reserves in the stubbles of drought stressed *L. perenne* at the end of a drought stress period. Both of which

463 could have contribute to increased growth rates observed in their study once rewetting had occurred. Also, drought-

464 induced shifts in plant phenology could lead to a shift in high productive stages, e.g. leading to peak growth rates not in

465 spring, but in summer (O'Toole and Cruz, 1980). With the data we collected throughout our experiment, we cannot clearly

466 identify the mechanisms behind the strong and consistent post-drought growth increase that extended even into the next

467 growing season. In the end, several biogeochemical and ecophysiological mechanisms might be responsible for the

468 overcompensation of growth following drought release.

469

470 ***4.5 Grass species and cultivars only slightly differed in drought resistance and recovery***

471 During the seasonal drought events the six tested grass species and cultivars showed a mostly universal response with

472 only slight and not consistent differences in their growth rate reductions. Post-drought legacy effects differed, however,

473 among the different grasses in the second year. *D. glomerata* and *P. pratensis* showed a high potential for recovery and

474 overcompensation after drought, while *L. multiflorum* generally showed the lowest recovery. Wang et al. (2007) found

475 that plant communities consisting of less productive species were more resistant to drought than plant communities

476 consisting of more productive species. The fact that inter-specific differences in the responses to the drought stress and

to the following rewetted post-drought period in our study were smaller than in other studies may be related to the fact
that all six tested grass species and cultivars belong to a relatively narrow functional group of productive fast-growing
grasses with high demands for mineral N in the soil. The availability of mineral N in the soil was found to be a key factor
for the response during as well as after drought for non-leguminous species (Hofer et al., 2017a, 2017b).

*4.6    Small to moderate impact of seasonal drought on annual ANPP*
Although the immediate effects of drought on growth rates were severe in all three seasons in our study, the overall effects
on total annual ANPP of -4 to -14% were only small to moderate compared to drought effects observed in other studies
(Gherardi and Sala, 2019; Wilcox et al., 2017; Wu et al., 2011) (Fig. 8). We also did not find any consistent effects of the
drought timing on annual ANPP, contrary to other studies (Denton et al., 2017; La Pierre et al., 2011; Nippert et al., 2006;
Petrie et al., 2018). This is likely a consequence of the small overall drought effects on annual ANPP in our study. The
small drought effects on annual ANPP that we report here are in line with Finn et al. (2018) and can be explained by the
high recovery of growth rates in the treatment plots following the drought release. This is in particular evident in the
spring treatment, where we observed on the one side the largest absolute reduction in growth in response to drought, but
at the same time also the strongest recovery after drought, leading to relatively small total drought effects on annual
ANPP. Because the fall drought treatment period lasted until the end of the vegetation period, the positive post-drought
legacy effects for this treatment were not included in the calculation of annual biomass production. Nevertheless, the fall
drought treatment in 2014 did also not strongly affect the annual ANPP. This is because the growth period affected by
the fall drought treatment, was the least productive part of the growing season, and, thus contributed only little to the
annual productivity.
The overall effect of drought on annual ANPP might also be small compared to other studies because our study
was conducted in highly productive grasslands that, according to best practice management, were harvested six times in
the growing season. The drought treatments occurred, however, only in two out of these six growth periods throughout
the growing season. In addition, the first sheltered growth period generally did not show a reduced growth rate (Fig. 4),
as soil water stress in this period was low (Fig. 2, Table 2). With the absence of negative legacy effects, the impact of the
immediate drought effect of one drought stressed growth period on annual NPP was therefore diluted by the five other
harvests of the vegetation period (Finn et al., 2018).
The majority of studies that have assessed the impact of drought on grassland productivity have either assessed
immediate drought effects, i.e. drought resistance (Bollig and Feller, 2014; Kahmen et al., 2005; Walter et al., 2012;
Wang et al., 2007), or have assessed the net effects of drought on annual NPP (Gherardi and Sala, 2019; Wilcox et al.,
2017; Wu et al., 2011). Our study highlights that it is important to also quantify immediate and post-drought effects –
even in the following growing season – if the causes of drought reduced annual productivity are to be understood.
Effects of drought on annual ANPP of grasslands have been shown to vary, depending on the severity of the
experienced drought (Vicca et al., 2012; Wilcox et al., 2017), ecosystem type (Byrne et al., 2013; Gherardi and Sala,
2019; Sala et al., 2015; Wilcox et al., 2017), the intensity of land use (Vogel et al., 2012; Walter et al., 2012), the plant
functional composition (Gherardi and Sala, 2015; Hofer et al., 2016, 2017a; Mackie et al., 2018), or the biodiversity of
an ecosystem (Haughey et al., 2018; Isbell et al., 2015; Kahmen et al., 2005; Wagg et al., 2017). Our study shows that
the timing of a drought event in the growing season is also crucial for the immediate effects of a drought on grassland
productivity. Importantly, however, our study also shows that strong positive legacy effects can occur after rewetting and
that these legacy effects are even important in spring of the next year. These effects can partially compensate the strong
immediate drought effects and lead to relatively small overall seasonal drought effects on annual ANPP.
Author contributions:
AK and AL planned, designed and supervised the research. CH and SEH performed the experiments. CH and MS
analyzed the data; CH wrote the manuscript. AK, AL and MS co-wrote the manuscript.
Acknowledgements
We thank Cornel Stutz and Rafael Gago for technical assistance in the field, as well as Florian Cueni for his support with
field work and sample processing. The Federal Office for Meteorology (MeteoSwiss) is kindly acknowledged for
providing access to meteorological data. We acknowledge financial support by the IDP BRIDGES project from the
European Union's Seventh Framework Programme (PITN-GA-643 2013; grant agreement no. 608422).

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

Tables
Table 1: Amount of precipitation fallen in the experiment and associated amount of excluded precipitation during the
sheltered drought periods in the years 2014 and 2015. Growing season precipitation refers to the time period between the
set-up of the shelters and the last harvest of each year.

| | | 2014 | | |
|---|---|---|---|---|
| annual precipitation (mm) | growing season precipitation (mm) | spring | summer | fall |
| | | excluded precipitation (mm) | | |
| 937.1 | 717.9 | 167.4 | 308.8 | 241.7 |
| | | excluded precipitation annually (%) | | |
| | | 17.9 | 33.0 | 25.8 |
| | | excluded precipitation in growing season (%) | | |
| | | 23.2 | 43.0 | 33.7 |
| | | 2015 | | |
| annual precipitation (mm) | growing season precipitation (mm) | spring | summer | fall |
| | | excluded precipitation (mm) | | |
| 801.9 | 648.5 | 296.9 | 204.7 | 149.9 |
| | | excluded precipitation annually (%) | | |
| | | 37.0 | 25.5 | 18.7 |
| | | excluded precipitation in growing season (%) | | |
| | | 45.8 | 31.6 | 23.1 |


Table 2: (a) Median of soil water potential, (b) stress intensity $I_S$ and (c) average air temperature during the two growth
periods of the drought treatments and the two post-drought growth periods as well as the corresponding periods of the
rain-fed control. Post-drought values of soil water potential, stress intensity $I_S$ and average air temperature are not
displayed (n.d.) as calculating these values for the long winter period between the end of the fall treatment and the spring
harvests has little meaning.

a)

| Growth period | Control | | | Treatment | | |
|---|---|---|---|---|---|---|
| | spring | summer | fall | spring | summer | fall |
| 2014 | | | | MPa | | |
| 1st drought | -0.03 | -0.41 | -0.01 | -0.09 | -0.72 | -0.73 |
| 2nd drought | -0.01 | -0.01 | -0.01 | -1.44 | -1.44 | -1.61 |
| 1st post-drought | -0.41 | -0.01 | n.d. | -1.1 | -0.05 | n.d. |
| 2nd post-drought | -0.01 | -0.01 | n.d. | -0.01 | -0.02 | n.d. |
| 2015 | | | | MPa | | |
| 1st drought | -0.01 | -0.02 | -0.14 | -0.08 | -0.45 | -0.85 |
| 2nd drought | -0.01 | -0.25 | -0.34 | -0.77 | -0.83 | -1.34 |
| 1st post-drought | -0.02 | -0.14 | n.d. | -0.57 | -0.73 | n.d. |
| 2nd post-drought | -0.25 | -0.34 | n.d. | -0.7 | -0.88 | n.d. |

b)

| Growth period | Control | | | Treatment | | |
|---|---|---|---|---|---|---|
| | spring | summer | fall | spring | summer | fall |
| 2014 | | | | | | |
| 1st drought | 0 | 8 | 0 | 1 | 13 | 3 |
| 2nd drought | 0 | 0 | 0 | 33 | 33 | 41 |
| 1st post-drought | 8 | 0 | n.d. | 24 | 9 | n.d. |
| 2nd post-drought | 0 | 0 | n.d. | 0 | 0 | n.d. |
| 2015 | | | | | | |
| 1st drought | 0 | 0 | 0 | 0 | 4 | 13 |
| 2nd drought | 0 | 0 | 1 | 14 | 4 | 34 |
| 1st post-drought | 0 | 0 | n.d. | 0 | 8 | n.d. |
| 2nd post-drought | 0 | 1 | n.d. | 14 | 13 | n.d. |

c)

| Growth period | Control | | | Treatment | | |
|---|---|---|---|---|---|---|
| | spring | summer | fall | spring | summer | fall |
| 2014 | | | | °C | | |
| 1st drought | 10.3 | 18.0 | 16.6 | 11.0 | 19.0 | 17.3 |
| 2nd drought | 10.9 | 18.0 | 15.2 | 11.5 | 18.7 | 15.8 |
| 1st post-drought | 18.0 | 16.6 | n.d. | 18.0 | 16.6 | n.d. |

| | | | | | | |
|---|---|---|---|---|---|---|
| 2nd post-drought | 18.0 | 15.2 | n.d. | 18.0 | 15.2 | n.d. |
| 2015 | | | °C | | | |
| 1st drought | 7.1 | 16.2 | 20.3 | 7.6 | 16.9 | 20.5 |
| 2nd drought | 13.3 | 22.7 | 13.0 | 14.4 | 23.7 | 13.5 |
| 1st post-drought | 16.2 | 20.3 | n.d. | 16.2 | 20.3 | n.d. |
| 2nd post-drought | 22.7 | 13.0 | n.d. | 22.7 | 13 | n.d. |


Table 3: Summary of analysis for the effects of season, drought treatment, grass species and cultivars (grass), and their
interactions on temperature-weighted growth rates (DMYTsum, natural log-transformed) from the second growth period
during drought. The inference (*F*- and *p*-values) refers to the fixed effects of the linear mixed model. $df_{num}$: degrees of
freedom term, $df_{den}$: degrees of freedom of error.

| Effect | $df_{num}$ | $df_{den}$ | 2014 | | 2015 | |
|---|---|---|---|---|---|---|
| | | | *F*-value | *p* | *F*-value | *p* |
| Season (spring, summer, fall) | 2 | 36 | 1051.1 | <0.001 | 2655.3 | <0.001 |
| Treatment (control vs. drought) | 1 | 72 | 341.9 | <0.001 | 642.9 | <0.001 |
| Grass | 5 | 72 | 9.4 | <0.001 | 14.2 | <0.001 |
| Season × Treatment | 2 | 72 | 25.9 | <0.001 | 366.2 | <0.001 |
| Season × Grass | 10 | 36 | 6.8 | <0.001 | 10.3 | <0.001 |
| Treatment × Grass | 5 | 72 | 2.9 | 0.018 | 2.0 | 0.094 |
| Season × Treatment × Grass | 10 | 72 | 3.3 | 0.001 | 3.4 | 0.001 |
| Marginal $R^2$ | | | 0.901 | | 0.965 | |
| Conditional $R^2$ | | | 0.917 | | 0.967 | |


Table 4: Summary of analysis for the effects of season, drought treatment, grass species and cultivars (grass), and their interactions on temperature-weighted growth rates (DMYTsum, natural log-transformed) from the second post-drought growth period. See Table 3 for additional explanation.

| Effect | $df_{num}$ | $df_{den}$ | 2014 | | 2015 | |
|---|---|---|---|---|---|---|
| | | | $F$-value | $p$ | $F$-value | $p$ |
| Season (spring, summer, fall) | 2 | 36 | 783.4 | <0.001 | 1428.6 | <0.001 |
| Treatment (control vs. drought) | 1 | 72 | 63.5 | <0.001 | 25.5 | <0.001 |
| Grass | 5 | 72 | 18.4 | <0.001 | 39.4 | <0.001 |
| Season × Treatment | 2 | 72 | 1.8 | 0.180 | 16.6 | <0.001 |
| Season × Grass | 10 | 36 | 15.7 | <0.001 | 9.6 | <0.001 |
| Treatment × Grass | 5 | 72 | 0.9 | 0.517 | 6.4 | <0.001 |
| Season × Treatment × Grass | 10 | 72 | 2.2 | 0.025 | 0.8 | 0.621 |
| Marginal $R^2$ | | | 0.810 | | 0.944 | |
| Conditional $R^2$ | | | 0.866 | | 0.946 | |





Figures

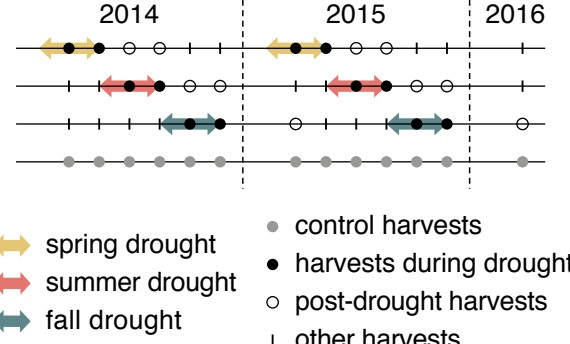


Fig. 1: Experimental design: The experiment lasted for two consecutive years (2014, 2015) with six evenly distributed
harvests in both years and one additional harvest in the beginning of 2016. Arrows indicate the duration of each drought
treatment (ten weeks). Each treatment was replicated four times for each of six grass species and cultivars.

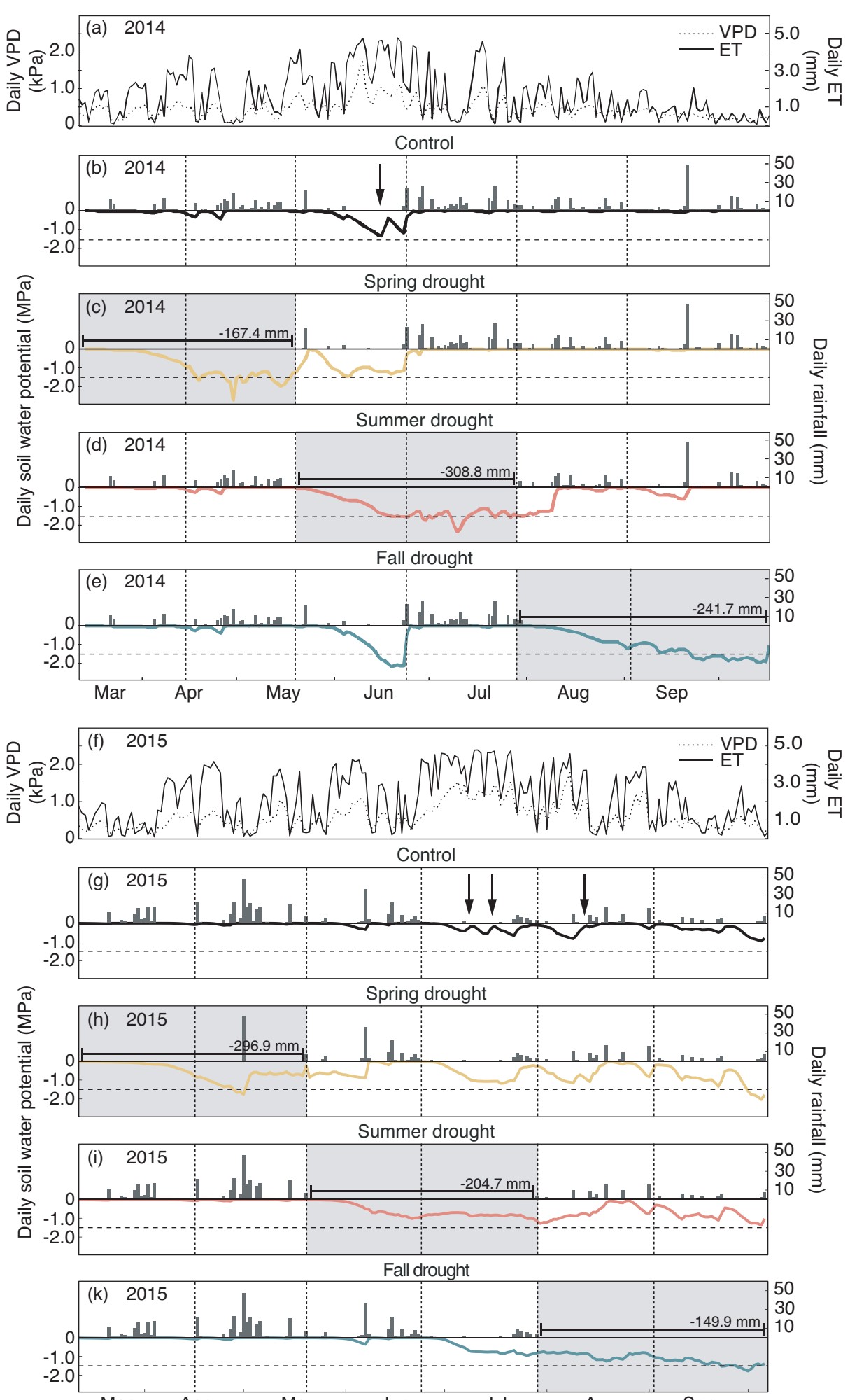

Fig. 2: (a, f) Daily evapotranspiration (ET) and vapor pressure deficit (VPD), (b-e, g-k) daily rainfall and soil water
potential ($\Psi_{Soil}$) in 10 cm depth over the growing seasons 2014 (a-e) and 2015 (f-k) for the control and drought treatment
(sensors per treatment: n=8). Grey shaded areas represent the experimental drought when rainfall was excluded. Dashed
horizontal line shows permanent wilting point ($\Psi_{Soil}$=-1.5MPa). Dashed vertical lines represent dates of harvest. Arrows
indicate watering events (in control plots only).

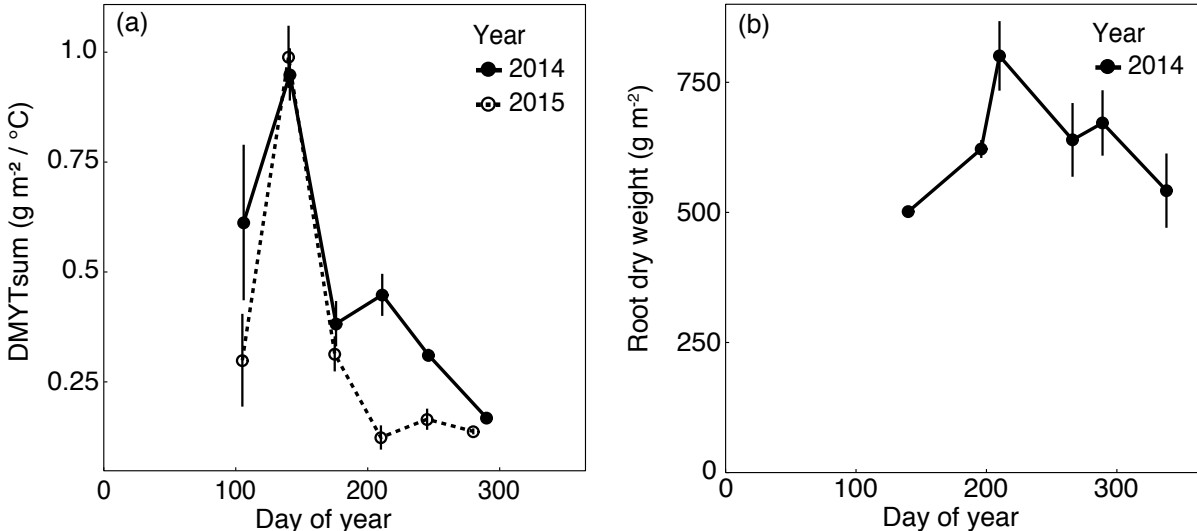


Fig. 3: (a) Temperature-weighted growth rates (DMYTsum) of aboveground biomass of rain-fed control plots in 2014
and 2015. Values displayed are the means across the six investigated grass species and cultivars (n=6, ± se). (b)
Belowground biomass of rain-fed control plots in 2014. Values displayed are the means across the four grasses *L. perenne*
early (LPe) and late (LPl) flowering and *D. glomerata* early (DGe) and late (DGl) flowering (n=4, ± se).

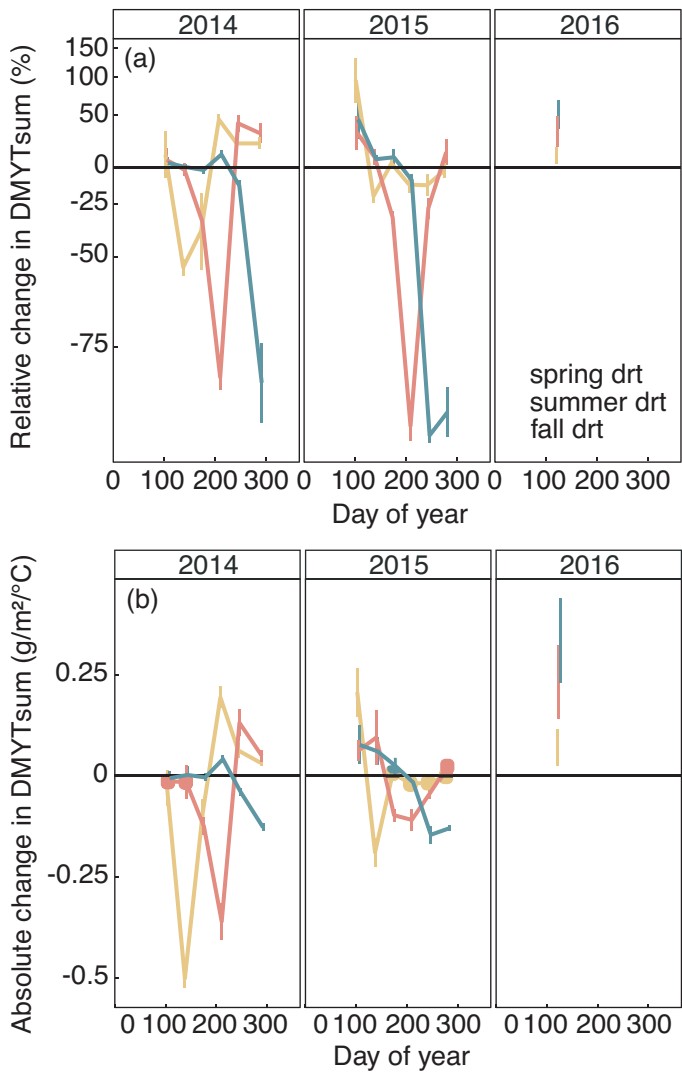

Fig. 4: (a) Relative (RCG) and (b) absolute (ACG) changes in temperature-weighted growth rates (DMYTsum) of the
respective drought (drt) treatment compared to the control (ctr) for 2014, 2015 and 2016. Values shown are means across
all six investigated grass species and cultivars (n=6, ± se). Values below the horizontal black line indicate reduced growth
compared to the control. Values above the line indicate an increase of growth.
RCG=100×(DMYTsum(drt)/DMYTsum(ctr))-1); displayed on log-scale); ACG=DMYTsum(drt)–DMYTsum(ctr).


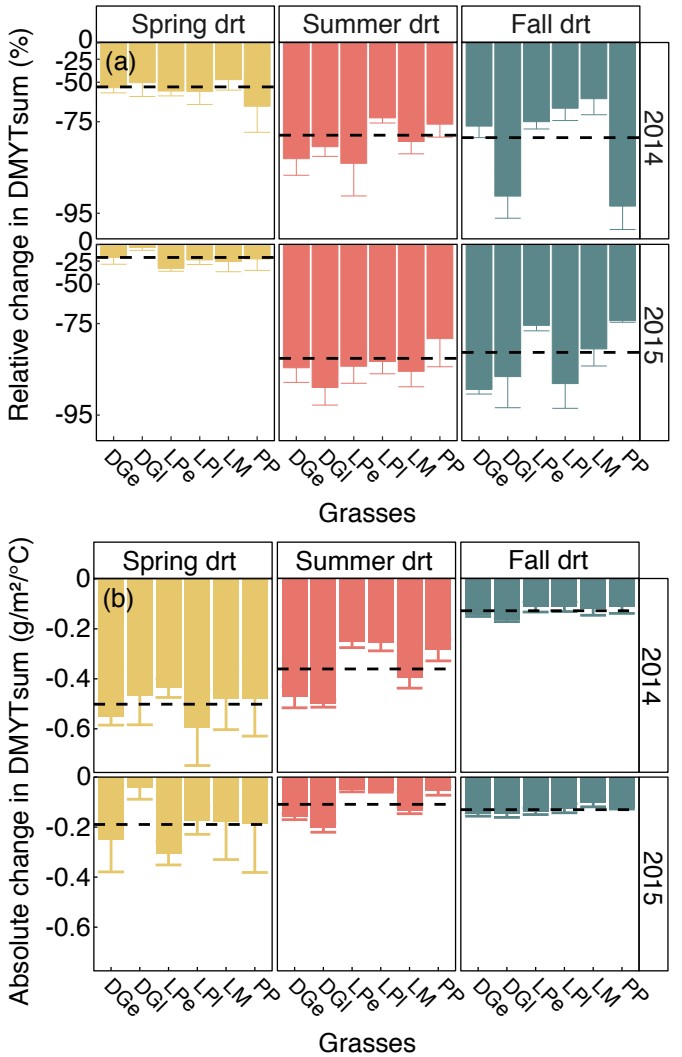

Fig. 5: (a) Relative (RCG) and (b) absolute (ACG) changes in temperature-weighted growth rates (DMYTsum) for the second growth period (weeks six to ten) of the respective drought (drt) treatment for 2014 and 2015 for the individual grasses. Values shown are means of four replicates per species and cultivar (n=4, ± se). Dashed black lines represent the means across all grasses. See Fig. 4 for additional explanation. The corresponding statistical analyses are shown in Table A1 in the Appendix.

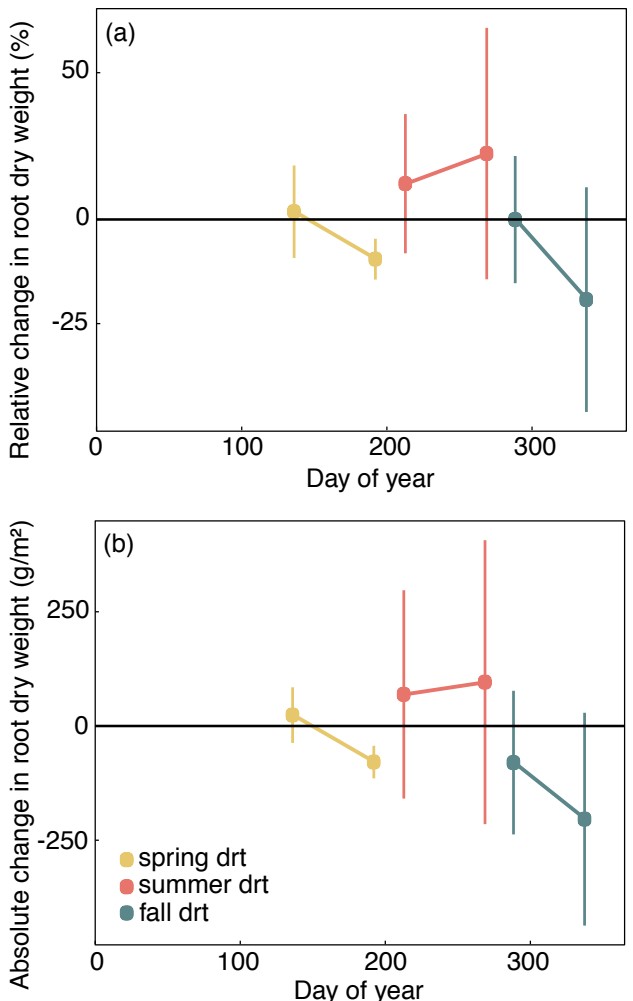

Fig. 6: (a) Relative and (b) absolute changes in root dry matter at the end of each drought treatment and after six to eight
weeks after drought-release in 2014. Values shown are means of four grasses of *L. perenne* (LPe and LPl) and *D.*
*glomerata* (DGe and DGl) each in four replicates (n=4, ± se).

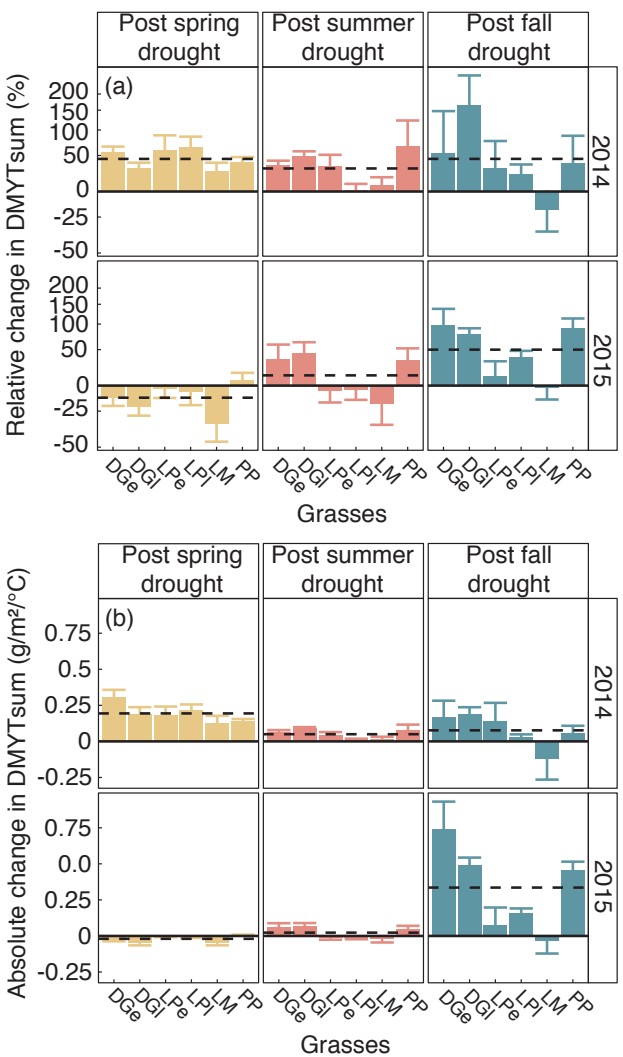


Fig. 7: (a) Relative (RCG) and (b) absolute (ACG) changes in temperature-weighted growth rates (DMYTsum) for the
second post-drought growth period (weeks six to ten) in 2014 and 2015 after the respective drought (drt) treatment for
the individual grasses. Values shown are means of four replicates (n=4, ± se). Post-drought growth period of the fall
drought treatment is the first growth period of the following year. See Fig. 4 for additional explanation. The corresponding
statistical analyses are shown in Table A1 in the Appendix.

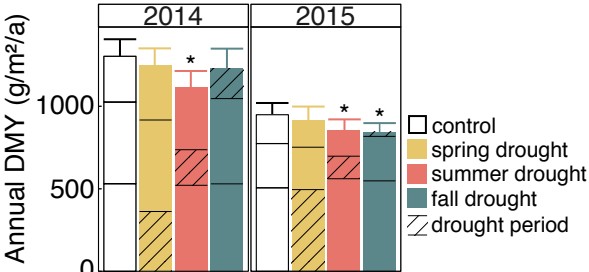

Fig. 8: Annual ANPP under rain-fed control and under the three seasonal drought treatments in the years 2014 and 2015.

Values shown are means across all six investigated grass species and cultivars (n=6, ± se). Bars in (a) are stacked

according to growth in spring (bottom part), summer and fall (top part). Significant differences to the control are marked

with * ($p<0.05$). The corresponding statistical analyses are shown in Table A2 in the Appendix.