# Peer review of "Timing of drought in the growing season and strong legacy effects"

_Biogeosciences, 2020_

## Referee Comment (RC1) · Anonymous Referee #1 · 22 May 2020

GENERAL COMMENTS

The authors performed a very interesting drought experiment on temperate grass monocultures, in which they assessed the effects of timing of drought on resistance, recovery and overall, annual (A)NPP. They found the lowest drought sensitivity in spring, and overcompensating post-drought growth which to a substantial extent cancelled out immediate impacts of drought on overall ANPP. By unravelling global change effects (drought) on components of ecosystem function (ANPP), the study matches the scope of the journal.

Overall, I am very enthusiastic about the design of the experiment with consideration of resistance, recovery and overall perturbation, the details and reporting of measurements performed, and the consideration of both absolute and relative effects. I found the text also very well structured and well written.

Despite my positive evaluation of the manuscript, I recommend major revisions at this stage because of three main issues to be resolved, and further comments below:

• Based on limited analyses on precipitation data and discussion on the course of soil water potential, the authors suggest that differences in spring vs summer vs fall soil moisture were likely not the main reason for the lower drought sensitivity in spring. While I tend to agree with the authors, they actually have all the necessary data to provide stronger, formal evidence that soil moisture stress was not particularly higher in summer and autumn than in spring. Based on the daily soil water potential, the field capacity and permanent wilting point, I recommend the authors to estimate daily soil moisture stress (Is), as explained in for example Vicca et al. (2012). In brief, with the approach proposed in that letter, plants or soil biota experience drought stress when soil moisture drops below a certain threshold, e.g. relative extractable water below 0.4. The amount below that threshold determines the severity of stress, and stress values for multiple days can be summed so you can get an idea of integrated soil moisture stress (e.g. for spring vs summer vs fall). I recommend the authors to calculate Is and report on their findings in the manuscript to strengthen their message, if confirmed. I also suggest to use Is in Table 2 and Figs. 4 and 9b. See also my references to this point in some of the specific comments below. Please avoid adding more display items, since four tables and nine figures is already at the higher end.

• While the design and procedures done in the experiment itself were well explained, the study still lacks reproducibility in the sense that no data nor R-script were provided in a supplement or link along with the manuscript. Ideally, both the data and a comprehensive script with the main code for statistical tests are uploaded. If it is not possible to make data publicly accessible, the authors need to explain that in a section "Data

availability" at the end of the manuscript. See also BG's data policy.

• Test statistics (F, df, P, ...) were not always presented along with the results in the text/tables/figures. If there is no place in figures to provide such information, please place tables in a supplement and refer to these in figure captions and in the Results section.

SPECIFIC COMMENTS

Line 14 – Here and throughout the paper, the authors refer to "resilience" to refer to post-drought recovery. In line with a proposal for standardized nomenclature and quantification of resilience proposed by Ingrisch & Bahn (2018), I suggest to replace "resilience" by "recovery" when specifically post-drought growth was meant. The overall "resilience", or the opposite, "perturbation", then combines both the "resistance" and "recovery" phases, resulting in e.g. the annual outcomes (see also Ingrisch et al., 2017 for an example).

Line 25 – From your experiment, you found that "(i) the resistance of growth rates in grasses to drought varies across the season and is positively correlated with growth rates in the control". While I agree that the first part of this claim will often be correct, I think there may in practice be many cases where drought resistance of growth – expressed either as absolute or relative values – will not correlate positively with control growth rates. For example, in an agricultural setting, N addition can promote plant growth under sufficient water supply (drought control), while it can exacerbate impacts of drought and thus reduce resistance (Wang et al., 2020). I suggest you either remove the second part of the sentence, or emphasize that it cannot be generalized.

Line 30 – Maybe the emphasis on Europe only is not needed in this paragraph. What about climate projections and ecosystem services of temperate grasslands elsewhere?

Line 65, 89, 90, ... – Suggestion to replace "resilience" by "recovery", see above.

Line 97 – Actually there were different cultivars of four grass species in total.

Line 166 – Were temperature sums calculated based on treatment-specific temperature measurements, as referred to in Table 2 and Line 130? So in other words, Tsum was slightly higher for the drought treatments than the controls? Please add this information.

Line 188 – Does PPT(ctr) include the few +20 mm watering events?

Line 188 – Please also quantify and compare S per unit change in soil moisture stress (Is) – see general comments.

Line 192 – You explain here the statistical analyses carried out. However, to improve reproducibility, I highly recommend you to (i) upload the data in the supplement/provide a link to the data (if allowed to share open-access), and (ii) provide a simple R-script with the code for the main analyses.

Line 194 – The word "regression" suggests that curves were fitted, while in fact only differences among levels in factors were assessed (i.e. ANOVA). While regression and ANOVA are statistically equivalent, I propose to replace "regression" by "models".

Line 198 – Plot was used as a random factor to take into account that the very same control plots were used for contrasting against different treatment plots in spring, summer and autumn. Should plot also be nested in grass and/or treatment? I did not think this through though, maybe it is redundant. Please comment.

Line 206 – Did $R^2$ refer to marginal or conditional $R^2$? If it does not apply here, please explain. Otherwise please provide both.

Line 212 – Unclear. You refer here to one-way ANOVA, after which two factors ($\sim$ two-way ANOVA) are mentioned. Please resolve.

Line 217 – Please mention and cite all R-packages used, e.g. for calculating and analyzing mixed-effects models, ...

Line 290 – Did you average first and then perform one-way ANOVA? See also my

comment on line 212. Please clarify.

Line 302 – "Drought (severity)" was defined here as precipitation reduction. Please check and report this also when expressing drought as a soil moisture index (Is). See general comments.

Line 305 – The statistical significance of the results (e.g. F- and P-values with df) is not given, here but also for other figures. If you think adding such details, even * symbols, would make figures confusing, then provide tables with statistics in a supplement and refer to those, mentioning on the significance in the Results section.

Line 308 – Suggestion to replace "resilience" by "resistance".

Line 320 – Not sure if you now have enough BNPP data to here and elsewhere claim that all changes in NPP will be equal to changes in ANPP. Perhaps it is safer to consistently refer to ANPP. As you mentioned in the section on root biomass (which I nevertheless recommend you to leave in the manuscript!), only the 0-14 cm layer was sampled. Maybe below more root biomass was produced during/after drought. Or would you suggest that this would in any case be negligible in magnitude compared to ANPP? Please comment.

Line 322 – See my comment on the Abstract about the positive correlation resistance ∼ control plot growth.

Line 327 – Please replace "climatic" by "meteorological" or "environmental". Climate rather refers to long-term statistics of the weather, not weather and soil moisture differences between two years.

Line 341 – Also here, it would be interesting to report on an integrated soil moisture stress index, besides/instead of median soil water potential. I then would expect Is to be significantly higher for 2015 than 2014.

Line 341 – I assume the median was taken because of the non-normal distribution of soil water potential data. However, to what extent is the median informative for any

reduction in growth? Or is it rather water potential values below a certain threshold that will affect growth?

Line 382 – Whether you consider differential soil moisture depletion among seasons as an artefact or not will depend on your point of view on the research questions. On the one hand, slower soil moisture depletion in spring than summer is something realistic that could be expected in many situations. On the other hand, it makes the unravelling of the mechanisms underlying lower drought sensitivity in spring more complicated. Please rephrase the "artefact" part.

Line 386 – Replace "herbs" by "forbs". Herbs include both forbs and graminoids. De Boeck et al. (2011) included only forbs in their experiment.

Line 388 – Here, I want to see reference to a formal test of differences in soil moisture stress. See also in the section with general comments. Note: it may be that soil moisture stress was significantly higher in summer than spring, but was still in the same order of magnitude. So this would not necessarily invalidate your suggestion that soil moisture alone could not explain the observations.

Line 397 – It seems that nowhere summarized data nor statistics were shown for root biomass per species/cultivar. Please provide such information in a supplement, and briefly refer to it in the Results section as well.

Line 426 – Besides N, also the availability of other nutrients like P and K can increase substantially after drought (see e.g. Van Sundert et al., 2020). These may have played a role as important as N, especially since N was added multiple times a year to minimize N limitation, whereas P, K and Mn were only added at the beginning of the growing seasons. Related to that, we could even speculate that P, K, ... were depleted because of harvests over the year, and perhaps a suboptimal P/K status contributed also to the increased drought sensitivity in summer and autumn. This last part is just a thought, I do not expect you to elaborate on this extensively in the manuscript, but please incorporate briefly the role and release of other nutrients in the text.

Line 453 – Please replace "resilience" by "recovery", see other comments.

Line 456 – Do not show statistics in the text of the Discussion section, unless absolutely necessary. Also, when P = n.s., I still prefer to see the actual P-value.

Line 484 – Refer here to more formal analyses, showing there was (almost) no soil moisture stress during this first growth period.

Line 710 – Better write "precipitation" instead of "rainfall". Maybe sometimes precipitation fell as snow or hail?

Table 2 – It would be interesting to see integrated soil moisture stress added to this table, or instead of median soil water potential.

Figure 4 – For this and other figures: I am not sure how easy or difficult it would be for a color-blind person to distinguish between the red and green. Consider using another color code.

Figure 4 – I am somewhat surprised to see that the + and - error bars in panel a have the same length, while the Y-axis was transformed. Is it because the transformation of the Y-axis was the same as the Y-variable in the analysis (e.g. ln)? And this was not the case for panel b then? Please explain or correct if necessary.

Figure 4 – Could you make this graph also for soil moisture stress, and then discuss whether change in growth followed change in stress.

Figure 5 – As indicated elsewhere, I am not a huge fan of this graph because correlation does not imply causation. While it is true that in your study, drought sensitivity of growth was highest when control growth was high, we cannot conclude in general that, where/when growth without water limitation is high, also drought resistance will be maximal.

Figure 7 – So did you first average the four plots per species, and then calculated mean plus se by combining the four species and taking n as 4? Or are these mixed model

outputs? This also applies to some other figures where multiple species were pooled. Please explain.

Figure 9 – Am I correctly interpreting that sensitivity did not significantly differ among seasons (no statistics shown)?

Figure 9 – I would like to see the sensitivity expressed per unit soil moisture drought stress, not only per mm of precipitation.

TECHNICAL CORRECTIONS Line 15 – Replace ", thus," by "eventually" or alike.

Line 91 – drought-stressed

Line 114 – Please remove "see".

Line 309 – Replace ", thus," by "eventually" or alike.

Line 309 – drought-induced reductions?

Line 445 – "Both could have contributed to increased growth rates (...)"

Line 459 – There is twice "the fact that" in this sentence. Please rewrite.

Line 500 – "lead to"?

REFERENCES

Ingrisch, J., & Bahn, M. (2018). Towards a comparable quantification of resilience. Trends in Ecology & Evolution, 33(4), 251–259.

Ingrisch, J., Karlowsky, S., Anadon-Rosell, A., Hasibeder, R., König, A., Augusti, A., Gleixner, G., & Bahn, M. (2018). Land use alters the drought responses of productivity and $CO_2$ fluxes in mountain grassland. Ecosystems, 21(4), 689–703.

Van Sundert, K., Brune, V., Bahn, M., Deutschmann, M., Hasibeder, R., Nijs, I., & Vicca, S. (2020). Post-drought rewetting triggers substantial K release and shifts in leaf stoichiometry in managed and abandoned mountain grasslands. Plant and Soil,

448, 353-368.

Vicca, S., Gilgen, A. K., Camino Serrano, M., Dreesen, F. E., Dukes, J. S., Estiarte, M., Gray, S. B., Guidolotti, G., Hoeppner, S. S., Leakey, A. D. B., Ogaya, R., Ort, D. R., Ostrogovic, M. Z., Rambal, S., Sardans, J., Schmitt, M., Siebers, M., van der Linden, L., van Straaten, O., & Granier, A. (2012). Urgent need for a common metric to make precipitation manipulation experiments comparable. New Phytologist, 195(3), 518–522.

Wang, Y., Huang, Y., Fu, W., Guo, W., Ren, N., Zhao, Y., & Ye, Y. (2020). Efficient physiological and nutrient use efficiency responses of maize leaves to drought stress under different field nitrogen conditions. Agronomy, 10(4), 523.
* * *

---

## Referee Comment (RC2) · Anonymous Referee #2 · 24 May 2020

The article presents the results of a seasonal drought manipulation experiment in Swiss grasses (six species) carried out in the growing seasons of 2014 and 2015. Specifically, results from three different rainfall exclusion strategies are presented: spring, summer, and fall rainfall exclusion subdivided in periods of 10 weeks each, as grass is harvested 6 times per year resulting in 6 growth periods. Nutrients were added to control and experimental plots. Beyond aboveground biomass harvest, root biomass, soil water potential, and meteorological conditions were also measured. The results show relatively minor difference across grass species. In relative terms, drought effects are

more pronounced for summer and fall treatments, while aboveground biomass is less affected by drought treatment during spring and root biomass is overall not affected. The study also shows that positive legacy effects can largely compensate for the reduction in aboveground biomass production during dry periods, leading to similar annual total aboveground biomass production between control and treatment scenarios.

The presented topic is interesting as there are not many seasonal drought studies, the experiment and results are clearly explained, and the manuscript is well organized. The fact that grass in treatment plots after the drought treatment outperformed the growth rates of the grasses in the controls for extended periods of time, suggesting a considerable resilience, is definitely an important result. However, while results are interesting, it is difficult to go beyond what has been observed and learn specific mechanisms (e.g., Line 378-380), as not many physiological variables are measured, e.g., the effects of drought on photosynthesis and stomatal conductance are not reported or maybe not observed (even though a mention to a manuscript in preparation is made). Additional physiological observations could have been useful to enter the debate of carbon source vs sink limitations in growth, which is very much active (e.g., Körner 2015). Potential explanation for the physiological mechanisms (e.g., osmoregulation) explaining the higher drought resistance of the investigated grasslands in spring and the capacity to compensate for growth after drought treatments could not be investigated in the article and are only speculated. Considering that any field or numerical experiments comes with limitations, I might be satisfied with these speculations.

What it is much less satisfying, is that the key question coming from data is left unanswered. Using the data in the article (see Fig. R1), we can clearly see that the ANPP sensitivity to growing season precipitation in the control scenarios is much, much larger than during drought treatments. This is not the first time, I see such type of "mechanistically unexplainable" behavior in field manipulation experiments. Now, the question is what is happening in "nature" that is not happening in the drought treatments? If the authors will add data from similar ecosystems (from literature) – something I would recommend to increase the outreach of the article - to the two observations, they will likely find a considerable sensitivity of grassland ANPP to precipitation for the natural rainfall regime. However, the sensitivity is very different in the treatments, even though at a lower "rainfall amount" sensitivity would be expected to even increase further rather than decrease (e.g., Huxman et al 2004). This result is somehow embedded in Fig. 9 and partially explained/discussed in 4.4 as a positive legacy effect. However, it is never presently as clearly as in Fig. R1 and of course, it leaves a big question mark on the representativeness of the entire study for real conditions. My explanation in such cases, it is typically that rainfall manipulation experiments have scale issues (lateral/vertical) that leads to such type of behavior. The authors have surely done their best to avoid any artifacts, but it remains the fact that the sensitivity they observe is completely different from the real sensitivity (but of course more years will be needed for a proper conclusion). This poses serious challenges on the extrapolation of the results to the real world. Some of the variability of ANPP can be ascribed to conditions other than precipitation, but it is difficult to find any convincing mechanistic explanation why ANPP sensitivity should be so different, and as this is unlikely what one observes in natural conditions, I am left with more doubts than answers.

Minor Comments

Line 66-77. There have been a number of publication from a drought experiment in a grassland in a similar environment near Innsbruck (e.g., Fuchslueger et al 2014; 2016, etc.), which can be relevant for this article.

Line 50. See also Paschalis et al 2020 for a recent analysis of model performance compared to rainfall manipulation experiments.

Line 83, 140-150 181-182. I know that it is very common to refer to grassland ANPP to the sum of harvested biomass throughout the year or the growing season. However, strictly speaking ANPP should be computed based on the continuous (flux) productivity allocated aboveground, i.e.., including also any turnover of biomass that might occur

between two harvests and also the change in biomass below the 7cm cut height. I think for grassland in Switzerland the difference might not be very significant but if the drought lead to some grass wilting and litter production, there could be some difference. Overall, I think it would be good to clearly mention that what is referred to as ANPP is not the "flux ANPP" but an estimated based on harvested biomass.

Line 131. Evapotranspiration is not a variable which is directly observed. How did you get the estimate? Which equation/method has been used to derive evapotranspiration?

Line 134-135. How many sensors were installed? How they were distributed? Could you be a bit more precise?

Line 136-137. While from a practical point of view, I agree with the authors, theoretically if transpiration among species differ also the soil water potential will differ especially in prolonged dry periods.

Line 227. Each different plant species or sometime even different individual of the same species will have a different "wilting point". I know that -1.5 MPa is (wrongly) a textbook reference number, but I would strongly suggest avoiding to indicate a "single" wilting point value.

Figure 4, 5 and 6. Maybe, all this information can be combined in a single Figure, especially Fig. 4 and 6.

Line 296-301. Please use (or not use) consistently the minus for a reduction in biomass. Now sometime is positive and sometime is negative.

Line 416-417. See also De Boeck et al 2018, who studied a not too dissimilar ecosystem even though at higher elevation.

Figure 1. I think this figure can be clearly improved adding a temporal axis with the proper dates and spacing between the harvests. Now, it is very conceptual and there is no reason as this is not a proposal but an experiment, which has been already carried out.

References

Fuchslueger L, Bahn M, Fritz K, Hasibeder R, Richter A. (2014). Experimental drought reduces the transfer of recently fixed plant carbon to soil microbes and alters the bacterial community composition in a mountain meadow. New Phytologist 201: 916–927

Fuchslueger, L., Bahn, M., Hasibeder, R., Kienzl, S., Fritz, K., Schmitt, M., Watzka, M. and Richter, A. (2016), Drought history affects grassland plant and microbial carbon turnover during and after a subsequent drought event. J Ecol, 104: 1453-1465. doi:10.1111/1365-2745.12593

Paschalis A., et al. (2020). Rainfall-manipulation experiments as simulated by terrestrial biosphere models: where do we stand? Global Change Biology 26(6), 3336-3355, doi.org/10.1111/gcb.15024

Huxman, T. E., et al. (2004), Convergence across biomes to a common rain use efficiency, Nature, 429, 651–654.

De Boeck HJ et al. (2018). Legacy Effects of Climate Extremes in Alpine Grassland. Front. Plant Sci., 30 October 2018 | https://doi.org/10.3389/fpls.2018.01586

Körner C. 2015. Paradigm shift in plant growth control. Current Opinion in Plant Biology 25: 107–114.

[Figure]

[Figure]

**Fig. 1.** Fig R1. Growing season ANPP vs growing season precipitation for the two control years (2014, 2015) and the six seasonal drought treatment carried out by the authors.

---

## Author Comment (AC1) · 26 Sep 2020

The authors performed a very interesting drought experiment on temperate grass monocultures, in which they assessed the effects of timing of drought on resistance, recovery and overall, annual (A)NPP. They found the lowest drought sensitivity in spring, and overcompensating post-drought growth which to a substantial extent cancelled out immediate impacts of drought on overall ANPP. By unravelling global change effects (drought) on components of ecosystem function (ANPP), the study matches the scope of the journal. Overall, I am very enthusiastic about the design of the experiment with

consideration of resistance, recovery and overall perturbation, the details and reporting of measurements performed, and the consideration of both absolute and relative effects. I found the text also very well structured and well written.

Despite my positive evaluation of the manuscript, I recommend major revisions at this stage because of three main issues to be resolved, and further comments below:

1) Based on limited analyses on precipitation data and discussion on the course of soil water potential, the authors suggest that differences in spring vs summer vs fall soil moisture were likely not the main reason for the lower drought sensitivity in spring. While I tend to agree with the authors, they actually have all the necessary data to provide stronger, formal evidence that soil moisture stress was not particularly higher in summer and autumn than in spring. Based on the daily soil water potential, the field capacity and permanent wilting point, I recommend the authors to estimate daily soil moisture stress (Is), as explained in for example Vicca et al. (2012). In brief, with the approach proposed in that letter, plants or soil biota experience drought stress when soil moisture drops below a certain threshold, e.g. relative extractable water below 0.4. The amount below that threshold determines the severity of stress, and stress values for multiple days can be summed so you can get an idea of integrated soil moisture stress (e.g. for spring vs summer vs fall). I recommend the authors to calculate Is and report on their findings in the manuscript to strengthen their message, if confirmed. I also suggest to use Is in Table 2 and Figs. 4 and 9b. See also my references to this point in some of the specific comments below. Please avoid adding more display items, since four tables and nine figures is already at the higher end.

Response: Thank you for this suggestion it confirms and helps to strengthen our interpretation of the results and, thus, the story of the paper. We followed the instructions of the referee and calculated Is according to Vicca et al. (2012) and present these data in table 2. As suspected by the referee, the resulting Is data confirm the cumulative soil water potential data that we report for the individual seasons and years. Our interpretation that "differences in spring vs summer vs fall soil moisture were likely not the main

reason for the lower drought sensitivity in spring" is thus confirmed by this additional analysis.

2) While the design and procedures done in the experiment itself were well explained, the study still lacks reproducibility in the sense that no data nor R-script were provided in a supplement or link along with the manuscript. Ideally, both the data and a comprehensive script with the main code for statistical tests are uploaded. If it is not possible to make data publicly accessible, the authors need to explain that in a section "Data availability" at the end of the manuscript. See also BG's data policy.

Response: All data and R scripts and now provided in a separate link.

3) Test statistics (F, df, P, ...) were not always presented along with the results in the text/tables/figures. If there is no place in figures to provide such information, please place tables in a supplement and refer to these in figure captions and in the Results section.

Response: We now added analytical statistics in additional tables in a supplement for figures 5, 6, 7 and 8 (updated figure numbers), and added test statistics (p-values) to the text. For figures 3 and 4 we did not add additional statistics because (i) these are two figures to give the reader the overview over the time course of plant growth over all the six harvests, (ii) the analyses of the individual key harvests are given in in tables 3 and 4, and (iii) compared to the huge differences in growth among the harvests, the standard errors are so small, that an additional table would not deliver additional information. Figures 5, 6, and 7 both comprise a panel for relative and absolute changes of the response variable. Here, all statistical analyses have been done with natural log transformed data, which was needed to meet the assumptions of the models. The analyses thus match panels a) in Figures 5, 6, and 7. Panels b) with the absolute changes complement this information and are given for a better understanding of the system, and the text in the Results contains descriptive means. Given this situation, it is neither needed nor common to provide further statistics, as

the relevant analyses are all done with the transformed data (which was indicated by the data itself). It would also be hard to find an appropriate parametric model for the absolute changes, given the distribution of values; and nonparametric methods are not available for multifactorial data structures. We hope that the referee kindly agrees to this strategy.

SPECIFIC COMMENTS

Line 14 – Here and throughout the paper, the authors refer to "resilience" to refer to post-drought recovery. In line with a proposal for standardized nomenclature and quantification of resilience proposed by Ingrisch Bahn (2018), I suggest to replace "resilience" by "recovery" when specifically post-drought growth was meant. The overall "resilience", or the opposite, "perturbation", then combines both the "resistance" and "recovery" phases, resulting in e.g. the annual outcomes (see also Ingrisch et al., 2017 for an example).

Response: We followed this advice and replaced "resilience" with "recovery" throughout the manuscript according to Ingritsch et al. 2017.

Line 25 – From your experiment, you found that "(i) the resistance of growth rates in grasses to drought varies across the season and is positively correlated with growth rates in the control". While I agree that the first part of this claim will often be correct, I think there may in practice be many cases where drought resistance of growth – expressed either as absolute or relative values – will not correlate positively with control growth rates. For example, in an agricultural setting, N addition can promote plant growth under sufficient water supply (drought control), while it can exacerbate impacts of drought and thus reduce resistance (Wang et al., 2020). I suggest you either remove the second part of the sentence, or emphasize that it cannot be generalized.

Response: We agree with the comment by the author that our study does not allow to conclude that highly productive grasslands are more drought resistant than low productivity grasslands. We do show, however, that the grass species and cultivars that

we investigated are more drought resistant in the phenological stage of highest productivity than in the other phenological stages where productivity is much lower (Fig. 3a). To clarify this, we re-wrote this statement in the abstract and in the discussion and deleted figure 5 as suggested by the referee.

Line 30 – Maybe the emphasis on Europe only is not needed in this paragraph. What about climate projections and ecosystem services of temperate grasslands elsewhere?

Response: We agree and therefore deleted the entire first paragraph.

Line 65, 89, 90, ... – Suggestion to replace "resilience" by "recovery", see above.

Response: Resilience was replaced by "recovery" here and throughout the manuscript.

Line 97 – Actually there were different cultivars of four grass species in total.

Response: We corrected this and now talk about four species of which two were grown in two cultivars.

Line 166 – Were temperature sums calculated based on treatment-specific temperature measurements, as referred to in Table 2 and Line 130? So in other words, Tsum was slightly higher for the drought treatments than the controls? Please add this information.

Response: Temperature sums were calculated based on treatment-specific temperature measurements. Thus, Tsum was slightly higher for drought treatments than for controls.

Line 188 – Does PPT(ctr) include the few +20 mm watering events?

Response: PPT(ctr) includes the +20 mm watering events.

Line 188 – Please also quantify and compare S per unit change in soil moisture stress (Is) – see general comments.

Response: We decided to delete sensitivity from figure 9 (independent of whether

calculated per mm precipitation reduction or per Is). The reason is, that it is related to another time span (annual) as well as to another basis of comparison (mm precipitation or Is) than all the other drought responses we present (individual harvest, absolute and relative loss of biomass). Due to the annual time span, it is an unfair comparison of the treatments spring, summer and autumn drought, because autumn drought has no "chance" for a compensation during recovery. In addition, changing time span and basis of comparison might lead to confusion and this new sensitivity detracts attention from the main message of the paper.

Line 192 – You explain here the statistical analyses carried out. However, to improve reproducibility, I highly recommend you to (i) upload the data in the supplement/provide a link to the data (if allowed to share open-access), and (ii) provide a simple R-script with the code for the main analyses.

Response: As explained above, we now provide these data. Since we describe the statistical analysis in detail in the manuscript, we prefer not to upload the R codes directly (which is also not required according to the journal policy).

Line 194 – The word "regression" suggests that curves were fitted, while in fact only differences among levels in factors were assessed (i.e. ANOVA). While regression and ANOVA are statistically equivalent, I propose to replace "regression" by "models".

Response: We replaced "regression" by "models".

Line 198 – Plot was used as a random factor to take into account that the very same control plots were used for contrasting against different treatment plots in spring, summer and autumn. Should plot also be nested in grass and/or treatment? I did not think this through though, maybe it is redundant. Please comment.

Response: The data matrix was coded so that each repeatedly measured plot was assigned an individual identifier. Under this condition, the lme() function in R correctly calculates the respective variance component given the structure of the fixed effects

"grass" and "treatment". It is neither needed nor appropriate to nest the plot variance within "grass" and/or "treatment".

Line 206 – Did R2 refer to marginal or conditional R2? If it does not apply here, please explain. Otherwise please provide both.

Response: The marginal and conditional R2 are now provided in all summary tables of the mixed-effects analyses.

Line 212 – Unclear. You refer here to one-way ANOVA, after which two factors (two-way ANOVA) are mentioned. Please resolve.

Response: This has been clarified (it was a two-way ANOVA), and we apologize for the typo.

Line 217 – Please mention and cite all R-packages used, e.g. for calculating and analyzing mixed-effects models,

Response: Done.

Line 290 – Did you average first and then perform one-way ANOVA? See also my comment on line 212. Please clarify.

Response: The ANOVA was performed on un-averaged data. This is now clarified in the new table A2 in the supplement.

Line 302 – "Drought (severity)" was defined here as precipitation reduction. Please check and report this also when expressing drought as a soil moisture index (Is). See general comments.

Response: As explained above, we decided to delete annual sensitivity from figure 9 altogether.

Line 305 – The statistical significance of the results (e.g. F- and P-values with df) is not given, here but also for other figures. If you think adding such details, even * symbols,

would make figures confusing, then provide tables with statistics in a supplement and refer to those, mentioning on the significance in the Results section.

Response: We agree and did add more details on statistics in tables in the supplement. More explanations are given above in the response to the general remark of R1 above.

Line 308 – Suggestion to replace "resilience" by "resistance".

Response: We guess it should read: replace "resilience" by "recovery". We changed throughout the manuscript.

Line 320 – Not sure if you now have enough BNPP data to here and elsewhere claim that all changes in NPP will be equal to changes in ANPP. Perhaps it is safer to consistently refer to ANPP. As you mentioned in the section on root biomass (which I nevertheless recommend you to leave in the manuscript!), only the 0-14 cm layer was sampled. Maybe below more root biomass was produced during/after drought. Or would you suggest that this would in any case be negligible in magnitude compared to ANPP? Please comment.

Response: We follow the suggestion of the referee and replaced NPP with ANPP throughout the entire manuscript.

Line 322 – See my comment on the Abstract about the positive correlation resistance. Control plot growth.

Response: This refers to the phonological stage (see chapter 4.2) of the highest growth rate which we now clarify in the text (and deleted figure 5). See more detailed response above (response to comment in line 25).

Line 327 – Please replace "climatic" by "meteorological" or "environmental". Climate rather refers to long-term statistics of the weather, not weather and soil moisture differences between two years.

Response: We replaced "climatic" by "meteorological" as suggested.

Line 341 – Also here, it would be interesting to report on an integrated soil moisture stress index, besides/instead of median soil water potential. I then would expect Is to be significantly higher for 2015 than 2014.

Response: In this section we report on soil water content and how it varied and not stress. We would thus like to not discuss the stress indicator here but we added Is to table 2 and discuss it in different parts of the discussion.

Line 341 – I assume the median was taken because of the non-normal distribution of soil water potential data. However, to what extent is the median informative for any reduction in growth? Or is it rather water potential values below a certain threshold that will affect growth?

Response: Stress is the product of duration of the stress and the intensity of the stress. Using the median was the best choice for us to combine both components of stress as good as possible in a single value. We decided against using arithmetic mean soil moisture values as it would potentially bias values towards a few extreme values and thus overemphasize soil moisture stress. Compared to the median of soil water potential, the metric of Is has the disadvantage that it is a yes / no response and does not take into consideration the increasing stress severity with soil water potential further decreasing over a the threshold of 0.4 MPa. Presenting both values (median of soil water potential and Is) has now the advantage that both can be seen by the reader. Interestingly, the values of the two variables (table 2) are highly correlated.

Line 382 – Whether you consider differential soil moisture depletion among seasons as an artefact or not will depend on your point of view on the research questions. On the one hand, slower soil moisture depletion in spring than summer is something realistic that could be expected in many situations. On the other hand, it makes the unravelling of the mechanisms underlying lower drought sensitivity in spring more complicated. Please rephrase the "artefact" part.

Response: We follow this suggestion by deleting "artefact. The sentence now reads:

"An alternative explanation for different immediate drought effects on growth rates throughout the growing season are different experimentally induced drought severities throughout a growing season"

Line 386 – Replace "herbs" by "forbs". Herbs include both forbs and graminoids. De Boeck et al. (2011) included only forbs in their experiment.

Response: We replaced "herbs" by "forbs".

Line 388 – Here, I want to see reference to a formal test of differences in soil moisture stress. See also in the section with general comments. Note: it may be that soil moisture stress was significantly higher in summer than spring, but was still in the same order of magnitude. So this would not necessarily invalidate your suggestion that soil moisture alone could not explain the observations.

Response: Due to not measuring soil moisture in all the replicates, statistics with significant levels is not possible. However, the metrics for stress severity presented in table 2a and 2b are impressively demonstrating that drought stress in summer was not more severe than in spring. Values for soil water potential median are for 2014 -1.44 MPa and -1.44 MPa for spring and summer respectively while they were for 2015 -0.77 MPa and -0.83 MPs. For Is the values were 33 and 33 for spring and summer in 2014 and 14 and 4 in 2015.

Line 397 – It seems that nowhere summarized data nor statistics were shown for root biomass per species/cultivar. Please provide such information in a supplement, and briefly refer to it in the Results section as well.

Response: The summary tables of these analyses are now provided in the supplement (Tab. A1) and we refer to these tables in the text.

Line 426 – Besides N, also the availability of other nutrients like P and K can increase substantially after drought (see e.g. Van Sundert et al., 2020). These may have played a role as important as N, especially since N was added multiple times a year to minimize N limitation, whereas P, K and Mn were only added at the beginning of the growing seasons. Related to that, we could even speculate that P, K, ... were depleted because of harvests over the year, and perhaps a suboptimal P/K status contributed also to the increased drought sensitivity in summer and autumn. This last part is just a thought, I do not expect you to elaborate on this extensively in the manuscript, but please incorporate briefly the role and release of other nutrients in the text.

Response: We now discuss the relevance of nutrients in more general terms and added the suggested reference van Sundert et al. 2020.

Line 453 – Please replace "resilience" by "recovery", see other comments.

Response: We replaced "resilience" by "recovery" here and throughout the manuscript.

Line 456 – Do not show statistics in the text of the Discussion section, unless absolutely necessary. Also, when P = n.s., I still prefer to see the actual P-value.

Response: We deleted the stats.

Line 484 – Refer here to more formal analyses, showing there was (almost) no soil moisture stress during this first growth period.

Response: We now refer to figure 2 and table 2 (ab) where we show this.

Line 710 – Better write "precipitation" instead of "rainfall". Maybe sometimes precipitation fell as snow or hail?

Response: We changed "rainfall" to"precipitation".

Table 2 – It would be interesting to see integrated soil moisture stress added to this table, or instead of median soil water potential.

Response: We added integrated values for Is.

Figure 4 – For this and other figures: I am not sure how easy or difficult it would be for a color-blind person to distinguish between the red and green. Consider using another

color code.

Response: We tested the colors with the tool "Color Oracle" to check if they are distinguishable for color-blind people. They are distinguishable for all 3 types of clolor.blindness.

Figure 4 – I am somewhat surprised to see that the + and - error bars in panel a have the same length, while the Y-axis was transformed. Is it because the transformation of the Y-axis was the same as the Y-variable in the analysis (e.g. ln)? And this was not the case for panel b then? Please explain or correct if necessary.

Response: Note that this is now figure 5, and the comment also applies to figures 6 and 7. Panel a) and b) have indeed not the same underlying scale. In panel a) the intervals have equal distances on the ln scale (with matches the parametric analyses, as suspected by the reviewer); correspondingly, the length of the error bars is the same in + and - direction. Next, these ln values are expressed in "percent change" (linear transformation from ln values, without changing the scale!) because this is more reader friendly, and it is then reasonable to specify a range of percent values in straight numbers (here e.g. 50, 100, 150, or -25, -50, -75). If now the intervals of these percent scales are evaluated, it turns out that the percent change of the error bars in + and − direction is not equivalent, although the plotted length is. Thus, the interpretation of errors fully matches the asymmetric errors bars, if the data (and the ln scale) would be back-transformed to linear scale. In the panel b) the scale is simply linear and means and standard errors are based on the absolute changes of the data without any transformation. Our approach is common practice, as e.g. can be seen in Figure 2 of "Schneider MK et al. (2014) Gains to species diversity in organically farmed fields are not propagated at the farm level. Nature Communications, 5."

Figure 4 – Could you make this graph also for soil moisture stress, and then discuss whether change in growth followed change in stress.

Response: We like this suggestion. In fact in a companion paper (Hahn et al. in prep),

where we report the physiological responses of the investigated grass species and cultivars to drought stress in spring, summer and fall we plot the physiological stress response over stress intensity experienced. In the current manuscript we prefer not to do this because we feel that the manuscript is already quite long with 9 figures and that an additional analysis would not really contribute to the overall findings we would like to report.

Figure 5 – As indicated elsewhere, I am not a huge fan of this graph because correlation does not imply causation. While it is true that in your study, drought sensitivity of growth was highest when control growth was high, we cannot conclude in general that, where/when growth without water limitation is high, also drought resistance will be maximal.

Response: We agree and deleted figure 5. Nevertheless, we would like to keep the message, that plants were most drought resistant during the most productive phenological stage in the growing season. However, the information that growth rate was much higher in the second regrowth than in the 4th and 6th regrowth (by a factor of 2 to 8 times higher!) can easily be depicted from figure 3. In addition, we now clarify in the text that this does not suggest that productive grasslands are more drought resistant than non-productive grasslands.

Figure 7 – So did you first average the four plots per species, and then calculated mean plus se by combining the four species and taking n as 4? Or are these mixed model outputs? This also applies to some other figures where multiple species were pooled. Please explain.

Response: Yes, we first averaged the replicates per species and then took n as 4 representing the different species. The means and SEs are calculated from raw data (as was done in all figures). Doing so, no specific indication is needed. If we would have presented model predictions, we would have indicated this with e.g. "predicted values from the model".

Figure 9 – Am I correctly interpreting that sensitivity did not significantly differ among seasons (no statistics shown)?

Response: We did delete annual sensitivity from figure 9 (old number). The main reason is that sensitivity during drought stress (figure 6, old number; table 3) should not be mixed up with annual sensitivity. In addition, annual sensitivity is not a fair comparison of the treatments because fall drought has no chance to compensate yield losses during recovery (as recovers happens only in spring next year).

Figure 9 – I would like to see the sensitivity expressed per unit soil moisture drought stress, not only per mm of precipitation.

Response: The same response as just above and as response to R1 comment to line 188.

TECHNICAL CORRECTIONS Line 15 – Replace ", thus," by "eventually" or alike. Line 91 – drought-stressed Line 114 – Please remove "see". Line 309 – Replace ", thus," by "eventually" or alike. Line 309 – drought-induced reductions? Line 445 – "Both could have contributed to increased growth rates (...)" Line 459 – There is twice "the fact that" in this sentence. Please rewrite. Line 500 – "lead to"?

Response: We incorporated the suggested corrections in the text.

---

## Author Comment (AC2) · 26 Sep 2020

The article presents the results of a seasonal drought manipulation experiment in Swiss grasses (six species) carried out in the growing seasons of 2014 and 2015. Specifically, results from three different rainfall exclusion strategies are presented: spring, summer, and fall rainfall exclusion subdivided in periods of 10 weeks each, as grass is harvested 6 times per year resulting in 6 growth periods. Nutrients were added to control and experimental plots. Beyond aboveground biomass harvest, root biomass, soil water potential, and meteorological conditions were also measured. The results show

relatively minor difference across grass species. In relative terms, drought effects are more pronounced for summer and fall treatments, while aboveground biomass is less affected by drought treatment during spring and root biomass is overall not affected. The study also shows that positive legacy effects can largely compensate for the reduction in aboveground biomass production during dry periods, leading to similar annual total aboveground biomass production between control and treatment scenarios.

The presented topic is interesting as there are not many seasonal drought studies, the experiment and results are clearly explained, and the manuscript is well organized. The fact that grass in treatment plots after the drought treatment outperformed the growth rates of the grasses in the controls for extended periods of time, suggesting a considerable resilience, is definitely an important result. However, while results are interesting, it is difficult to go beyond what has been observed and learn specific mechanisms (e.g., Line 378-380), as not many physiological variables are measured, e.g., the effects of drought on photosynthesis and stomatal conductance are not reported or maybe not observed (even though a mention to a manuscript in preparation is made). Additional physiological observations could have been useful to enter the debate of carbon source vs sink limitations in growth, which is very much active (e.g., Körner 2015). Potential explanation for the physiological mechanisms (e.g., osmoregulation) explaining the higher drought resistance of the investigated grasslands in spring and the capacity to compensate for growth after drought treatments could not be investigated in the article and are only speculated. Considering that any field or numerical experiments comes with limitations, I might be satisfied with these speculations.

Response: We thank the referee for this overall very positive evaluation. We have assessed ecophysiological variables in four out of the six species/cultivars (conductance, pre-dawn and midday water potential). These data will be presented in a different manuscript that is currently in the final stages of preparation. It was a strategic decision not to include physiological data in the current manuscript but to focus on the reported biomass patterns. We agree, however, that the reported patterns alone only

allow to speculate about the mechanisms. These will then be discussed in the second manuscript. Given the wealth of data that we present (biomass data from six harvests from 192 plots from two growing seasons), we did not want to overload this paper and are convinced that the reported patterns are yet interesting and valuable.

What it is much less satisfying, is that the key question coming from data is left unanswered. Using the data in the article (see Fig. R1), we can clearly see that the ANPP sensitivity to growing season precipitation in the control scenarios is much, much larger than during drought treatments. This is not the first time, I see such type of "mechanistically unexplainable" behavior in field manipulation experiments. Now, the question is what is happening in "nature" that is not happening in the drought treatments? If the authors will add data from similar ecosystems (from literature) – something I would recommend to increase the outreach of the article - to the two observations, they will likely find a considerable sensitivity of grassland ANPP to precipitation for the natural rainfall regime. However, the sensitivity is very different in the treatments, even though at a lower "rainfall amount" sensitivity would be expected to even increase further rather than decrease (e.g., Huxman et al 2004). This result is somehow embedded in Fig. 9 and partially explained/discussed in 4.4 as a positive legacy effect. However, it is never presently as clearly as in Fig. R1 and of course, it leaves a big question mark on the representativeness of the entire study for real conditions. My explanation in such cases, it is typically that rainfall manipulation experiments have scale issues (lateral/vertical) that leads to such type of behavior. The authors have surely done their best to avoid any artifacts, but it remains the fact that the sensitivity they observe is completely different from the real sensitivity (but of course more years will be needed for a proper conclusion). This poses serious challenges on the extrapolation of the results to the real world. Some of the variability of ANPP can be ascribed to conditions other than precipitation, but it is difficult to find any convincing mechanistic explanation why ANPP sensitivity should be so different, and as this is unlikely what one observes in natural conditions, I am left with more doubts than answers.

Response: The referee raises an important point. We did, however, consider this when planning the experiment. The main reasons for this discrepancy most probably are:

(i) Between the two years not only precipitation differs, but potentially a lot of other abiotic (e.g. temperature, frost events) and biotic (e.g. diseases, soil microbial activity, age of the sward) differ. This is the reason why experiments to study drought effects need to compare drought stress treatments with a rainfed control under otherwise exactly the same conditions.

(ii) The timing of a lack of precipitation is crucial. This can nicely be demonstrated by the response of the grasses to 100

Consequently, we are convinced that such a comparison of whole growing season precipitation differences among years have only very limited validity to explain drought response.

We are also convinced that our treatments did not induce important artefacts. The shelters were open on all four sides and on top to guarantee good airflow. Gutters guiding the water away from the plots and not harvested plot borders of 75cm width can guarantee, that lateral water flow did not affect the studied centre of the plots (as do the soil water potential measurements).

We can help to explain the reason why the response to the whole growth precipitation difference looks so big in the figure of the reviewer. Firstly, during the 10 weeks of the spring treatment in 2015 precipitation was exceptionally high +130 mm higher than in 2014 (Table 1). In contrast, the summer and fall periods 2015 were exceptionally dry with -195 mm less precipitation than in 2014. This had a huge effect on growth as it was in a crucial time period and because the soil water deficits lasted very long (Figure 2). There are now two effects that make the annual comparison so impressively responsive: First, the difference between the +130 mm and the -195 mm looks like a very small difference in growing season precipitation and second, the effect on plant growth was huge because the soil water deficit lasted so long (about 20 weeks), what

is much longer than the second 5 weeks of our drought treatments.

Minor Comments

Line 66-77. There have been a number of publication from a drought experiment in a grassland in a similar environment near Innsbruck (e.g., Fuchslueger et al 2014; 2016, etc.), which can be relevant for this article.

Response: We now include these references.

Line 50. See also Paschalis et al 2020 for a recent analysis of model performance compared to rainfall manipulation experiments.

Response: We now include this reference.

Line 83, 140-150 181-182. I know that it is very common to refer to grassland ANPP to the sum of harvested biomass throughout the year or the growing season. However, strictly speaking ANPP should be computed based on the continuous (flux) productivity allocated aboveground, i.e.., including also any turnover of biomass that might occur between two harvests and also the change in biomass below the 7cm cut height. I think for grassland in Switzerland the difference might not be very significant but if the drought lead to some grass wilting and litter production, there could be some difference. Overall, I think it would be good to clearly mention that what is referred to as ANPP is not the "flux ANPP" but an estimated based on harvested biomass.

Response: Thank you for this advice. We now define the reported ANPP values as "standing above-ground biomass".

Line 131. Evapotranspiration is not a variable which is directly observed. How did you get the estimate? Which equation/method has been used to derive evapotranspiration?

Response: We now include a reference in the text to clearly indicate the origin of these data.

Line 134-135. How many sensors were installed? How they were distributed? Could

you be a bit more precise?

Response: We installed 32 sensors that were randomly distributed among the plots. We now clarify this in the text.

Line 136-137. While from a practical point of view, I agree with the authors, theoretically if transpiration among species differ also the soil water potential will differ especially in prolonged dry periods.

Response: We agree. Nevertheless, in a previous study we assessed soil moisture decline at the same site in monocultures of the same species assessed here and found no differences. We therefore feel that the transpiration is comparable across plots with different species. We also compared the soil water potential values obtained from 32 plots in this study and found no species-specific effects suggesting mostly identical transpiration rates.

Line 227. Each different plant species or sometime even different individual of the same species will have a different "wilting point". I know that -1.5 MPa is (wrongly) a textbook reference number, but I would strongly suggest avoiding to indicate a "single" wilting point value.

Response: This comment is correct. Please be aware, however, that we use the permanent wilting point to assess from where onwards a treatment is experiencing critically low levels of soil moisture. While using a single threshold for all species/cultivars might add some uncertainty for across species comparisons, we would like to emphasize that our main focus is on the across season comparison of drought effects. A slight under or over estimation of the permanent wilting point would thus merely introduce a systematic effort that should not influence the overall outcome of our analysis.

Figure 4, 5 and 6. Maybe, all this information can be combined in a single Figure, especially Fig. 4 and 6.

Response: We actually had larger figures with more panels in a previous version of

the manuscript. In the end we decided against this as the figures as they are right now already contain quite a lot of data (already 6 and 12 panels). We are afraid that expanding the figures further would make them more difficult to comprehend. Figure 5 was deleted to be more concise. In addition, we do not see how figures 4 and 6 could be combined, because figure 4 gives all harvest (but averages the six grasses) while figure 6 gives all six grasses but only one single harvest.

Line 296-301. Please use (or not use) consistently the minus for a reduction in biomass. Now sometime is positive and sometime is negative.

Response: We followed this suggestion and now consistently use the minus symbol for negative changes.

Line 416-417. See also De Boeck et al 2018, who studied a not too dissimilar ecosystem even though at higher elevation.

Response: We included De Boeck et al 2018 in the text.

Figure 1. I think this figure can be clearly improved adding a temporal axis with the proper dates and spacing between the harvests. Now, it is very conceptual and there is no reason as this is not a proposal but an experiment, which has been already carried out.

Response: As the time span between the harvests was always five weeks the spacing in the figure is actually the proper temporal spacing during the experiment. We feel that a time axis and dates would not add substantial information as this is given in figure 2.

---

## Author Response (AR2)

General comments:

1)      Analyses of drought stress index "Is" were performed and reported on, but it seems to me that "Is" was not calculated correctly. Please see the specific comments.

Reply: The referee is correct, that volumetric water content would be the appropriate variable to use for the calculation of Is. In the design of our experiment we installed sensors measuring soil water potential rather than volumetric water content since we believe that this is the variable describing plant-available soil moisture in a much more comparable way than measurements of volumetric soil moisture. Please also note that we installed 42 sensors across our experiment, which we think is an exceptionally large number compared to other experiments that we are aware of. Unfortunately, we did not measure pF curves on our site since there is only so much that you can do and we figured that this was not necessary given our detailed assessment of soil water potential throughout the experiment. (There was no intention to calculate Is when we initiated this large experiment).

We agree, however, with the referee that it would be nice to have additional explanatory variables (as for most every experiment). This is why we originally agreed to the referee's feedback and calculated values for Is. It now turns out that we do not have the appropriate input variables (vol soil moisture) to calculate Is. Since we do not have the capacity to collect additional data (pF curves) at the site, the only solution we see would be to use a standardized pF curve to calculate Is. Such pF asked our coworkers if we could utilize their pF curves to estimate volumetric water content for our experiment. The agreed but advised us strongly not to use these curves, given the heterogeneity of the soils and the potential errors that this would introduce to our data. In particular, parameters in the existing pF curves from nearby sites change very much with depth and reliable results can therefore not be expected.

Given the uncertainty associated with these calculations, we would kindly like to request if we can ignore the suggestion of the referee and calculate Is values. We feel that the very large number of soil moisture sensors that we had installed delivered soil water potential data that are very well suited to determine the drought severity experienced by the vegetation throughout the experimental periods. We cannot take the responsibility to add ls to the manuscript under the given circumstances.

2)      According to your replies, the dataset and R script would be given, but I cannot find any link to these files in the manuscript. If I somehow overlooked the link, please refer to where it is shown, or otherwise insert the link. Also, in general, I highly recommend to refer in your author replies to where in the manuscript changes have been made, e.g. Lines X -Y, and if relevant enough (not simply a technical correction, ...), also quote these lines.

Reply: We are glad to provide the raw data of our experiment. We have formatted the data- and metafiles so that they are ready for upload. Unfortunately, we did not find any option on the web interface of the journal that allows us to upload the files. It would be kind if you could let us know where we to upload the respective files.

3)      Reviewer #2 made an excellent remark on variation in inter-annual ANPP vs intra-annual variation in ANPP across treatments. I at first sight more or less agree with your replies, but it seems you did not really incorporate your answers in the manuscript. Or if you did, please refer to where this information has been inserted in the manuscript, see also my previous comment. While Rev#2's remark may not be about the primary research questions, I agree on its importance and it should be briefly addressed in at least a few lines or a paragraph of Discussion.

Reply: We address now the point about "drought effects of sheltered treatments versus drought effect if the two seasons are compared" in two ways in the manuscript. First (lines 229-234), we changed the description of the weather conditions of the season 2015 compared to 2014. Now we present results of precipitation and evapotranspiration for the crucial time period "second half of the growing season" when the two years strongly differed. With this we are convinced to correct the wrong impression our earlier "over the whole growing season" description gave. These "over the whole growing season" values masked the severity of drought in 2015 considerably, because the first third of 2015 was really wet.

Second (lines 506-513), we now discuss in the manuscript the main reason of the generally low annual yields in 2015, which is the long lasting drought (stress during three regrowth cycles) compared to the sheltered drought treatments (stress during one regrowth cycle). As the yields during spring growth were comparable among the two years, it is quite obvious that the exceptionally dry conditions in summer and fall of the year 2015 are the main driver of lower yield.

In addition:
We think that these strongly differing weather conditions are a strength of our two-year study. We put this forward now in the revised manuscript by adding:
(lines 329-330) This pattern seems to be robust as it occurred in two years with strongly differing weather conditions.
(lines 374-375) This pattern was robust as it occurred in both years even though the years differed strongly in their weather conditions.

Specific comments:

See general comments.

Technical corrections:

Line 145 – "Eg" should be "Eq."
Reply: We corrected the spelling mistake.
Line 147 – Symbols for field capacity vs wilting point were confused.
Reply: Indeed. We corrected the mistake.
Line 355 – Suggestion to remove "the" from "the Is".
Reply: Since Is calculations were omitted from the manuscript, the correction is redundant.
Line 526 – The reference list was given twice in this manuscript version.
Reply: We corrected the mistake.
Table 2 – Some "Is" values were highlighted in violet. Please remove.
Reply: Since Is calculations were omitted from the manuscript, the correction is redundant.

[revised manuscript text omitted]